# A Mean Field Theory of Quantized Deep Networks: The Quantization-Depth Trade-Off

**Yaniv Blumenfeld**
Technion, Israel
yanivblm6@gmail.com

**Dar Gilboa**
Columbia University
dargilboa@gmail.com

**Daniel Soudry**
Technion, Israel
daniel.soudry@gmail.com

## Abstract

Reducing the precision of weights and activation functions in neural network training, with minimal impact on performance, is essential for the deployment of these models in resource-constrained environments. We apply mean field techniques to networks with quantized activations in order to evaluate the degree to which quantization degrades signal propagation at initialization. We derive initialization schemes which maximize signal propagation in such networks, and suggest why this is helpful for generalization. Building on these results, we obtain a closed form implicit equation for $L_{\max}$, the maximal trainable depth (and hence model capacity), given $N$, the number of quantization levels in the activation function. Solving this equation numerically, we obtain asymptotically: $L_{\max} \propto N^{1.82}$.

## 1 Introduction

As neural networks are increasingly trained and deployed on-device in settings with memory and space constraints [12, 5], a better understanding of the trade-offs involved in the choice of architecture and training procedure are gaining in importance. One widely used method to conserve resources is the quantization (discretization) of the weights and/or activation functions during training [6, 25, 13, 3]. When choosing a quantized architecture, it is natural to expect depth to increase the flexibility of the model class, yet choosing a deeper architecture can make the training process more difficult. Additionally, due to resource constraints, when using a quantized activation function whose image is a finite set of size $N$, one would like to choose the smallest possible $N$ such that the model is trainable and performance is minimally affected. There is a trade-off here between the capacity of the network which depends on its depth and the ability to train it efficiently on the one hand — and the parsimony of the activation function used on the other.

We quantify this trade-off between capacity/trainability and the degree of quantization by an analysis of wide neural networks at initialization. This is achieved by studying *signal propagation* in deep quantized networks, using techniques introduced in [24, 26] that have been applied to numerous architectures. Signal propagation will refer to the propagation of correlations between inputs into the hidden states of a deep network. Additionally, we consider the dynamics of training in this regime and the effect of signal propagation on the change in generalization error during training.

In this paper,

- We suggest (section 3.2) that if the signal propagation conditions do not hold, generalization error in early stages of training should not decrease at a typical test point, potentially explaining the empirically observed benefit of signal propagation to generalization. This is done using an analysis of learning dynamics in wide neural networks, and corroborated numerically.
- We obtain (section 4.2) initialization schemes that maximize signal propagation in certain classes of feed-forward networks with quantized activations.

- Combining these results, we obtain an expression for the trade-off between the quantization level and the maximal trainable depth of the network (eq. 18), in terms of the depth scale of signal propagation. We experimentally corroborate these predictions (Figure 3).

## 2 Related work

Several works have shown that training a 16 bit numerical precision is sufficient for most machine learning applications [10, 7], with little to no cost to model accuracy. Since, many more aggressive quantization schemes were suggested [13, 18, 22, 21], ranging from the extreme usage of 1-bit at representations and math operations [25, 6], to a more conservative usage of 8-bits [3, 28], all in effort to minimize the computational cost with minimal loss to model accuracy. Theoretically, it is well known that a small amount of imprecision can significantly degrade the representational capacity of a model. For example, an infinite precision recurrent neural network can simulate a universal Turing machine [27]. However, any numerical imprecision reduces the representational power of these models to that of finite automata [19]. In this paper, we focus on the effects of quantization on training. So far, these effects are typically quantified empirically, though some theoretical work has been done in this direction (e.g. [17, 1, 36, 33]).

Signal propagation in wide neural networks has been the subject of recent work for fully-connected [24, 26, 23, 31], convolutional [30] and recurrent architectures [4, 8]. These works study the evolution of covariances between the hidden states of the network and the stability of the gradients. These depend only on the leading moments of the weight distributions and the nonlinearities at the infinite width limit, greatly simplifying analysis. They identify critical initialization schemes that allow training of very deep networks (or recurrent networks on long time sequence tasks) without performing costly hyperparameter searches. While the analytical results in these works assume that the layer widths are taken to infinity sequentially (which we will refer to this as the *sequential* limit), the predictions prove predictive when applied to networks with layers of equal width once the width is typically of the order of hundreds of neurons. For fully connected networks it was also shown using an application of the Central Limit Theorem for exchangeable random variables that the asymptotic behavior at infinite width is independent of the order of limits [20].

## 3 Preliminaries: the mean field approach

### 3.1 Signal propagation in feed-forward networks

We now review the analysis of signal propagation in feed-forward networks performed in [24, 26]. The network function $f : \mathbb{R}^{n_0} \to \mathbb{R}^{n_{L+1}}$ is given by

$$
\begin{aligned}
\phi(\alpha^{(0)}(x)) &= x \\
\alpha^{(l)}(x) &= W^{(l)}\phi(\alpha^{(l-1)}(x)) + b^{(l)} \quad l = 1, ..., L \\
f(x) &= \alpha^{(L+1)}(x)
\end{aligned}
\tag{1}
$$

for input $x \in \mathbb{R}^{n_0}$, weight matrices $W^{(l)} \in \mathbb{R}^{n_l \times n_{l-1}}$ and nonlinearity $\phi : \mathbb{R} \to \mathbb{R}$. The weights are initialized using $W_{ij}^{(l)} \sim \mathcal{N}(0, \frac{\sigma_w^2}{n^{(l-1)}}), b_i^{(l)} \sim \mathcal{N}(0, \sigma_b^2)$ so that the variance of the neurons at every layer is independent of the layer widths [1].

According to Theorem 4 in [20], under a mild condition on the activation function that is satisfied by any saturating nonlinearity, the *pre-activations* $\alpha^{(l)}(x)$ converge in distribution to a multivariate Gaussian as the layer widths $n_1, ..., n_L$ are taken to infinity in any order (with $n_0, n_{L+1}$ finite) [2]. In the physics literature the approximation obtained by taking this limit is known as the *mean field approximation*.

The covariance of this Gaussian at a given layer is then obtained by the recursive formula

$$
\mathbb{E}\alpha_i^{(l)}(x)\alpha_j^{(l)}(x') = \mathbb{E}\sum_{k,k'=1}^{n_{l-1}} W_{ik}^{(l)}W_{jk'}^{(l)}\phi(\alpha_k^{(l-1)}(x))\phi(\alpha_{k'}^{(l-1)}(x')) + b_i^{(l)}b_j^{(l)}
$$
$$
= \left[\sigma_w^2\mathbb{E}\phi(\alpha_1^{(l-1)}(x))\phi(\alpha_1^{(l-1)}(x')) + \sigma_b^2\right]\delta_{ij}. \tag{2}
$$

Omitting the dependence on the inputs $x, x'$ in the RHS below, we define

$$
\begin{pmatrix} \mathbb{E}\alpha_i^{(l)}(x)\alpha_i^{(l)}(x) & \mathbb{E}\alpha_i^{(l)}(x)\alpha_i^{(l)}(x') \\ \mathbb{E}\alpha_i^{(l)}(x)\alpha_i^{(l)}(x') & \mathbb{E}\alpha_i^{(l)}(x')\alpha_i^{(l)}(x') \end{pmatrix} = Q^{(l)}\begin{pmatrix} 1 & C^{(l)} \\ C^{(l)} & 1 \end{pmatrix} = \Sigma(Q^{(l)}, C^{(l)}). \tag{3}
$$

Combining eqs. 2 and 3 we obtain the following two-dimensional dynamical system:

$$
\begin{pmatrix} Q^{(l)} \\ C^{(l)} \end{pmatrix} = \begin{pmatrix} \sigma_w^2 \underset{u\sim\mathcal{N}(0,Q^{(l-1)})}{\mathbb{E}}\phi^2(u) + \sigma_b^2 \\ \frac{1}{Q^{(l-1)}}\left[\sigma_w^2 \underset{(u_1,u_2)\sim\mathcal{N}(0,\Sigma(Q^{(l-1)},C^{(l-1)}))}{\mathbb{E}}\phi(u_1)\phi(u_2) + \sigma_b^2\right] \end{pmatrix} \equiv \mathcal{M}\left[\begin{pmatrix} Q^{(l-1)} \\ C^{(l-1)} \end{pmatrix}\right],
$$
$$
\tag{4}
$$

where $\mathcal{M}$ depends on the nonlinearity and the initialization hyperparameters $\sigma_w^2, \sigma_b^2$ and the initial conditions $(Q^{(0)}, C^{(0)})^T$ depend also on $x, x'$. See Figure 1 for a visualization of the covariance propagation.

Once the above dynamical system converges to a fixed point $(Q^*, C^*)$ or at least approaches it to within numerical precision, information about the initial conditions is lost. As argued in [26], this is detrimental to learning as inputs in different classes can no longer be distinguished in terms of the network output (assuming the fixed point $C^*$ is independent of $C^{(0)}$, see Lemma 1). The convergence rate to the fixed point can be obtained by linearizing the dynamics around it. This can be done for the two dimensional system as a whole, yet in [26] it was also shown that, for any monotonically increasing nonlinearity, convergence of this linearized dynamical system in the direction $C^{(l)}$ cannot be faster than convergence in the $Q^{(l)}$ direction, and thus studying convergence can be reduced to the simpler one dimensional system $C^{(l)} = \mathcal{M}_{Q^*}(C^{(l-1)})$ that is obtained by assuming $Q^{(l)}$ has already converged, as assumption we review in appendix K. The convergence rate is given by the following known results of [26, 8] which we recapitulate for completeness:

**Lemma 1.** [26, 8] Defining $\Sigma(Q, C) = Q\begin{pmatrix} 1 & C \\ C & 1 \end{pmatrix}$ for $Q \geq 0, C \in [-1, 1]$ the dynamical system

$$
\mathcal{M}_{Q^*}(C) = \frac{1}{Q^*}\left[\sigma_w^2 \underset{(u_1,u_2)\sim\mathcal{N}(0,\Sigma(Q^*,C))}{\mathbb{E}}\phi(u_1)\phi(u_2) + \sigma_b^2\right] \tag{5}
$$

when linearized around a fixed point $C^*$, converges at a rate

$$
\chi = \left.\frac{\partial \mathcal{M}_{Q^*}(C)}{\partial C}\right|_{C^*} = \sigma_w^2 \underset{(u_a,u_b)\sim\mathcal{N}(0,\Sigma(Q^*,C^*))}{\mathbb{E}}\phi'(u_1)\phi'(u_2). \tag{6}
$$

Additionally, $\mathcal{M}_{Q^*}(C)$ has at most one stable fixed point in the range $[0,1]$ for any choice of $\phi$ such that $\phi$ is odd or $\phi''$ is non-negative.

Proof: See Appendix A.

We subsequently drop the subscript in $\mathcal{M}_{Q^*}(C)$ to lighten notation. The corresponding time scale of convergence in the linearized regime is

$$
\xi = -\frac{1}{\log\chi}. \tag{7}
$$

$\chi$ depends on the initialization hyperparameters and choice of nonlinearity, and it follows from the considerations above that signal propagation from the inputs to the outputs of a deep network would be facilitated by a choice of $\chi$ such that $\xi$ diverges, which occurs as $\chi$ approaches 1 from below. Indeed, as observed empirically across multiple architectures and tasks [30, 4, 8, 31], up to a constant factor $\xi$ typically gives the maximal depth up to which a network is trainable. These calculations motivate initialization schemes that satisfy:

$$
\chi = 1
$$

in order to train very deep networks. We will show shortly that this condition is unattainable for a large class of quantized activation functions. [3]

The analysis of forward signal propagation in the sense described above in networks with continuous activations is related to the stability of the gradients as well [26]. The connection is obtained by relating the rate of convergence $\chi$ to the first moment of the state-to-state Jacobian

$$J = \lim_{l \to \infty} \frac{\partial \hat{\alpha}^{(l)}}{\partial \hat{\alpha}^{(l-1)}}.$$
(8)

Taking all the layer widths to be equal to $n$, the first moment is given by

$$m_{JJ^T} = \frac{1}{n} \mathbb{E} \text{tr} \left( JJ^T \right).$$
(9)

Since high powers of this matrix will appear in the gradient, controlling its spectrum can prevent the gradient from exploding or vanishing. In the case of quantized activations, however, the relationship between the Jacobian and the convergence rate $\chi$ no longer holds since the gradients vanish almost surely and modified weight update schemes such as the Straight-Through Estimator (STE) [11, 13] are used instead. However, one can define a modified Jacobian $J_{\text{STE}}$ that takes the modified update scheme into account and control its moments instead.

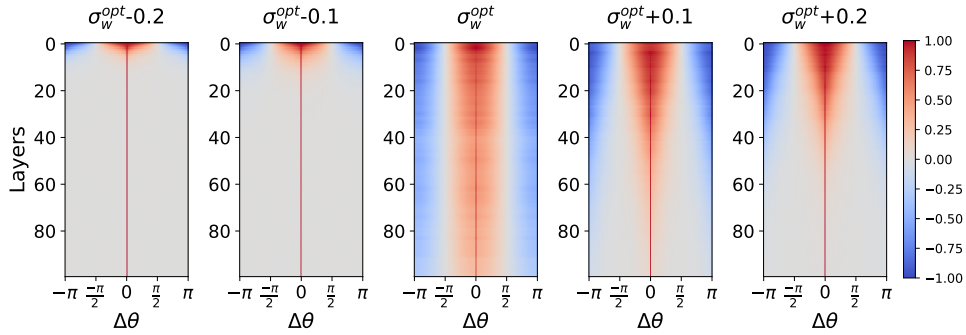

Figure 1: Propagation of empirical covariance between hidden states at different layers, in quantized feed-forward networks with $N = 16$, varying the standard deviation of the weights $\sigma_w$. $\Delta\theta$ is the angle between two normalized inputs. Signal propagation is maximized when $\sigma_w = \sigma_w^{\text{opt}}$, and degrades as $\sigma_w$ deviates from it.

## 3.2 Signal propagation may improve generalization

The argument that a network will be untrainable if signals cannot propagate from the inputs to the loss, corresponding to the rapid convergence of the dynamical system eq. 4, has empirical support across numerous architectures. A choice of initialization hyperparameters that facilitates signal propagation has also been shown to lead to slight improvements in generalization error, yet understanding of this was beyond the scope of the existing analysis. Indeed, there is also empirical evidence that when training very deep networks it is only the generalization error that is impacted adversely but the training error is not [30]. Additionally, one may wonder whether a deep network could still be trainable despite a lack of signal propagation. On the one hand, rapid convergence of the correlation map between the pre-activations is equivalent to the distance between $f(x), f(x')$ converging to a value that is independent of the distance between $x, x'$. On the other, since deep networks can fit random inputs and labels [34] this convergence may not impede training. .

To understand the effect of signal propagation on generalization, we consider the dynamics of learning for wide, deep neural networks in the setting studied in [14, 16]. We note that this setting introduces an unconventional scaling of the weights. Despite this, it should be a good approximation for the early stages of learning in networks with standard initialization, as long as the weights do not change too much from their initial values. In this regime, the function implemented by the network evolves linearly in time, with the dynamics determined by the Neural Tangent Kernel (NTK). We argue that

rapid convergence of eq. 4 in deep networks implies that the error at a typical test point should not decrease during training since the resulting form of the NTK will be independent of the label of the test point. Conversely, this effect will be mitigated with a choice of hyperparameters that maximizes signal propagation, which could explain the beneficial effect on generalization error that is observed empirically. We provide details and empirical evidence in support of this claim for networks with both quantized and continuous activation functions in Appendix M.

## 4    Mean field theory of signal propagation with quantized activations

In this section, we will explore the effects of using a quantized activation function on signal propagation in feed-forward networks. We will start by developing the mean field equations for a sign activations and then consider more general activation function, and establish a theory that predicts the relationship between the number of quantization states, the initialization parameters, and the feed-forward network depth.

### 4.1    Warm-up: sign activations

We begin by considering signal propagation in the network in eq. 1 with $\phi(x) = \text{sign}(x)$. Substituting $\phi(x) = \text{sign}(x)$, $\phi'(u) = 2\delta(u)$ in eqs. 4 and 6 gives

$$Q^* = \sigma_w^2 + \sigma_b^2, \ \ \chi = 4\sigma_w^2 \mathop{\mathbb{E}}_{(u_1, u_2) \sim \mathcal{N}(0, \Sigma(Q^*, C^*))} \delta(u_1)\delta(u_2). \tag{10}$$

As shown in Appendix C, we obtain

$$\chi = \frac{2\sigma_w^2}{\pi \left( \sigma_w^2 + \sigma_b^2 \right) \sqrt{1 - (C^*)^2}} \tag{11}$$

$$\mathcal{M}(C) = \frac{\frac{2\sigma_w^2}{\pi} \arcsin(C) + \sigma_b^2}{\sigma_w^2 + \sigma_b^2}. \tag{12}$$

The closed form expressions 11 and 12, which are not available for more complex architectures, expose the main challenge to signal propagation. It is clear from these expressions that the derivative of $\mathcal{M}(C)$ diverges at 1, and since $\mathcal{M}(C)$ is differentiable and convex, it can have no stable fixed point in $[0, 1]$ that satisfies the signal propagation condition $\chi = 1$. In fact, as we show in Appendix L.1 that the maximal value of $\chi$ for this architecture is achievable when $\sigma_b = 0$, and is bounded from above by $\chi_{\max} = \frac{2}{\pi}$ for all choices of the initialization hyperparameters. The corresponding depth scale is bounded by $\xi_{\max} < 3$.

Additionally, one may wonder if using stochastic binary activations [13] might improve signal propagation. In Appendix D we show this is not the case: we consider a stochastic rounding quantization scheme and show that stochastic rounding can only further decrease the signal propagation depth scale.

### 4.2    General quantized activations

We consider a general activation function $\phi_N : \mathbb{R} \to S$, where $S$ is a finite set of real numbers of size $|S| = N$. To obtain a flexible class of non-decreasing functions of this form, we define

$$\phi_N(x) = A + \sum_{i=1}^{N-1} H\left( x - g_i \right) h_i, \tag{13}$$

where $A \in \mathbb{R}, \forall i \in \{1, 2, ..., N-1\}, g_i \in \mathbb{R}, h_i \in \mathbb{R}_{>0}$, and $H : \mathbb{R} \to \mathbb{R}$ is the Heaviside function. This activation function can be thought of as a "stairs" function, going from the minimum state of $A$ to the maximum state $A + \sum_{i=1}^{N-1} h_i$, over $N - 1$ stairs, with stair $i$ located at an offset $g_i$ with a height $h_i$. We will assume that the offsets $g_i$ are ordered, for simplicity. The development of the mean field equations for this activation function is located in appendix E, where we find that:

$$\widehat{Q}^{(l)} = \sum_{i=1}^{N-1} \sum_{j=1}^{N-1} h_i h_j \Phi\left( -\frac{\max(g_i, g_j)}{\sqrt{Q^{(l)}}} \right) \Phi\left( \frac{\min(g_i, g_j)}{\sqrt{Q^{(l)}}} \right), \ \ Q^{(l+1)} = \sigma_w^2 \widehat{Q}^{(l)} + \sigma_b^2 \tag{14}$$

$$\chi = \frac{\sigma_w^2}{2\pi Q^* \sqrt{1 - (C^*)^2}} \sum_{i=1}^{N-1} \sum_{j=1}^{N-1} h_i h_j \exp\left[-\frac{g_i^2 - 2C^* g_i g_j + g_j^2}{2Q^* \left(1 - (C^*)^2\right)}\right], \tag{15}$$

where $\Phi$ is the gaussian CDF and $\widehat{Q}^{(l)}$ is the hidden state covariance, as explained in appendix B. This expression diverges as $C^* \to 1$ since all the summands are non-negative and the diagonal ones simplify to $\frac{\sigma_w^2 h_i^2}{2\pi Q^* \sqrt{1-(C^*)^2}} \exp\left[-\frac{g_i^2}{2Q^*(1+C^*)}\right]$. Since $\mathcal{M}(C)$ is convex (see Lemma 1), we find that as in the case of sign activation, $\chi = 1$ is not achievable for any choice of a quantized activation function.

To optimize the signal propagation for any given number of states, we would like to find the parameters that will bring the fixed point slope $\chi$ as close as possible to 1. For simplicity, we will henceforth use the initialization $\sigma_b = 0$, which is quite common [9]. Empirical evidence in appendix F suggest that using $\sigma_b > 0$ is sub-optimal, which is not very surprising, given our similar (exact) results for sign activation. For $\sigma_b = 0$, $C = 0$ becomes a fixed point. We eliminate eq. 15 direct dependency on $Q^*$, by defining *normalized offsets* $\tilde{g} \equiv \frac{g}{\sqrt{Q^*}}$. By moving to normalized offsets, substituting $C^* = 0$ and the remaining $Q^*$ by eq. 14, our expression for the fixed point slope becomes:

$$\chi = \frac{\sum_{i=1}^{N-1} \sum_{j=1}^{N-1} \frac{1}{2\pi} \exp\left[-\frac{1}{2}\left(\tilde{g}_i^2 + \tilde{g}_j^2\right)\right] h_i h_j}{\sum_{i=1}^{N-1} \sum_{j=1}^{N-1} \Phi\left(-\max(\tilde{g}_i, \tilde{g}_j)\right) \Phi\left(\min(\tilde{g}_i, \tilde{g}_j)\right) h_i h_j} \tag{16}$$

Eq. 16 provides us with way to determine the quality of any quantized activation function in regard to signal propagation, without concerning ourselves with the initialization parameters, that will only have a linear effect on the offsets. Since the normalized offsets are sufficient to determine $\widehat{Q}, Q$, using eq. 15, moving from normalized offsets to actual offsets becomes trivial.

To measure the relation between the number of states and depth scale, we will use eq. 16 over a limited set of constant-spaced activations, where we choose $A < 0, \forall i \in \{1, .., N-1\}, h_i = \text{const.}$ and the offsets are evenly spaced and centered around zero, with $D$ defined as the distance between two sequential offsets so that $g_i = D\left(i - \frac{N}{2}\right)$, and $\tilde{D}$ defined as $\tilde{D} = \frac{D}{\sqrt{Q^*}}$. We view this configuration as the most obvious selection of activation function, where the 'stairs' are evenly spaced between the minimal and the maximal state. Using eq. 16 on an activation in this set, we get:

$$\chi = \frac{\sum_{i \in K} \sum_{j \in K} \frac{1}{2\pi} \exp\left[-\frac{1}{2}\left(i^2 + j^2\right)\tilde{D}^2\right]}{\sum_{i \in K} \sum_{j \in K} \Phi\left(-\max(i, j)\tilde{D}\right) \Phi\left(\min(i, j)\tilde{D}\right)} \tag{17}$$

when $K = \left\{k - \frac{N}{2} | \forall k \in \mathbb{N}, k < N\right\}$. A numeric analysis using of eq. 17 is presented in figure 2, and reveals a clear logarithmic relation between the level of quantization to the optimal fixed point slope, and the normalized spacing required to reach this optimal configuration. By extrapolating the numerical results, as seen in the right panels of Fig. 2, we find a good approximations for the the maximal achievable slope for any quantization level $\chi_{\max}(N)$ and the corresponding normalized spacing $D_{\text{opt}}(N)$. Using those extrapolated values, we predict the depth-scale of a quantized, feed-forward network to be:

$$\xi_N = -\frac{1}{\log(\chi_{\max}(N))} \simeq -\frac{1}{\log(1 - e^{0.71}(N+1)^{-1.82})} \simeq \frac{1}{2}(N+1)^{1.82}. \tag{18}$$

where the latter approximation is valid for large $N$. While the depth scale in eq. 18 is applicable to uniformly spaced quantized activations, numerical results presented in Appendix G suggest that using more complex activations with the same quantization level will not produce better results.

In their work regarding mean field theory of convolutional neural networks, [30] shows that the dynamics of hidden-layer's correlations in CNNs decouple into independently evolving Fourier modes that evolves near the fixed point, each with a corresponding fixed-point-slope of $\chi_c \lambda_i$, with $\chi_c$ depending the initialization hyperparameters and equivalent to the fixed point slope as calculated for fully connected networks, and $\lambda_i \leq 1$ being a frequency dependant modifier corresponding to mod $i$. While the exact dynamics in this case may depend on the decomposition of the input to Fourier mods, it is apparent that the maximal depth-scale of each mod can not exceed the depth-scale calculated for the fully-connected case, and thus our upper limit on the number of layers holds for the case of CNNs.

Similarly, following [4] and [8], our results can be easily extended to single layer RNNs, LSTMs and GRUS, in which case the limitation applies to the timescale of the network memory.

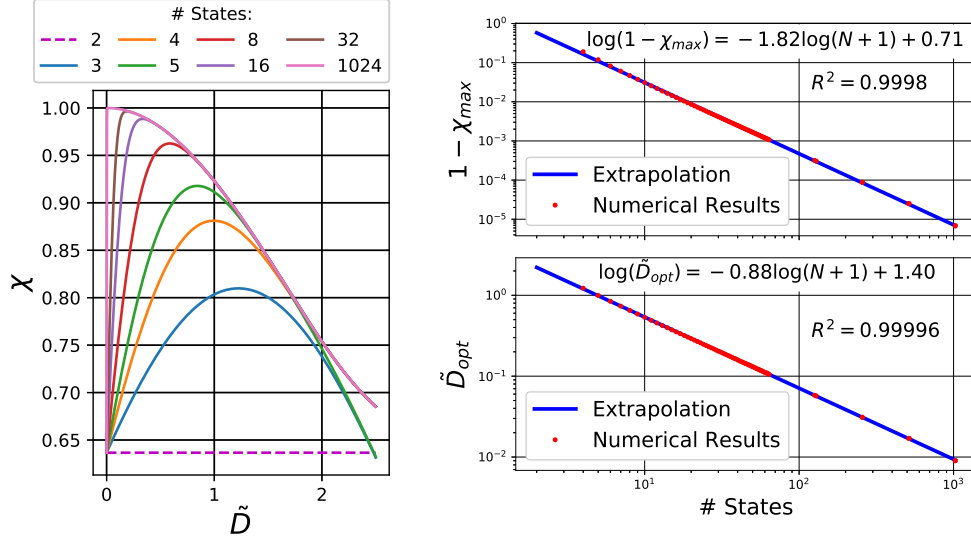

Figure 2: Numerical analysis of the covariance propagation fixed point slope for quantized activation functions. *Left:* The convergence rate in eq. 17 of the covariances of the hidden states as a function of the normalized spacing between offsets $\tilde{D}$ for activations with different levels of quantization $N$. *Top Right:* The difference between 1 and maximal achievable convergence rate $\chi_{\max}$ as a function of $N$. *Bottom Right:* The normalized spacing between states $\tilde{D}$ corresponding to $\chi_{\max}$ as a function of $N$. We find that the dependence of $1 - \chi_{\max}$ on $N$ is approximated well by a power law.

## 5    Experimental results

To visualize the covariance propagation in eq. 2 we reconstruct an experiment presented in [24], and apply it to untrained quantized neural networks. We consider a neural network with $L = 100$ fully-connected layers, all of width $n = 1000$. We draw two orthonormal vectors $u^0, u^1 \in \mathbb{R}^{1000}$ and generate the 1 dimensional manifold $U = \left\{ u_i = \sqrt{Q_s^*} \left( u^0 \cos(\theta) + u^1 \sin(\theta) \right) \middle| i \in \{0, \frac{1}{r}, \frac{2}{r}, .., \frac{r-1}{r}\}, \theta = 2\pi i \right\}$, where $r = 500$ is the number of samples, and $Q_s^*$ is the fixed point, calculated numerically. After initializing the neural network, we use the manifold values as inputs to the neural network and measure the covariance in all hidden layers. We then plot in Figure 1 the empirical covariance of the hidden states as a function of the difference in the angle $\theta$ of their corresponding inputs. The reason for multiplying the initial values by $\sqrt{Q_s^*}$ is so we can isolate the convergence of the off-diagonal correlations from that of the diagonal.

To test the predictions of the theory, we have constructed a similar experiment to the one described in [26], training neural networks of varying depths over the MNIST dataset. We study how the maximal trainable depth of a quantized activation fully-connected network depends on the weight variance $\sigma_w^2$ and the number of states in the activation function $N$. For our quantized activations, we used the constant-spaced activations we have analyzed in section 4.2:

$$\phi_N(x) = -1 + \sum_{i=1}^{N-1} \frac{2}{N-1} H \left( x - \frac{2}{N-1} \left( i - \frac{N}{2} \right) \right),$$

which describes an activation function with a distance of $D = \frac{2}{N-1}$ between offsets, and with states ranging between -1 and 1.

To find the best initialization parameters for each activation function, we first used eq. 14 to compute $\widehat{Q}^*$ assuming our normalized spacing $\frac{D}{\sqrt{Q^*}}$ is optimized ($\tilde{D}_{\text{opt}}$, computed using the linear regression parameters of Figure 2 bottom right panel). Then, we picked $\sigma_b = 0$, $\sigma_w = \frac{1}{\sqrt{\widehat{Q}^*}} \frac{D}{\tilde{D}_{\text{opt}}}$, and thus ensured that the normalized offsets are indeed optimal. Gradients are computed using the

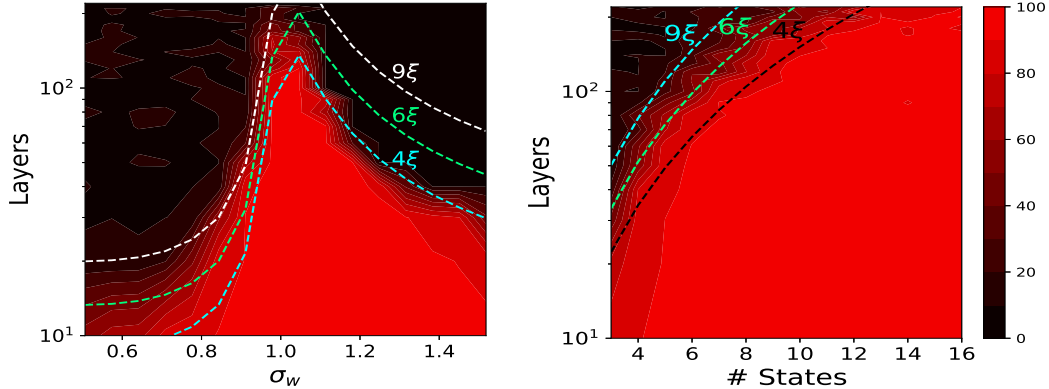

Figure 3: Test accuracy of feed-forward networks of different depth with quantized activation functions trained on MNIST classification after 1600 training steps, compared with the theoretical depth scale predictions (eq. 7). Up to a constant factor, the theoretical depth scale predicts the phase transition between regimes where a network is trainable and one where training fails. *Left:* Networks with a 10 states activation function and different values of the weight variance. *Right:* Networks with different quantization levels (number of states), with variances adjusted to allow optimal signal propagation.

Straight-Through Estimator (STE) [13]:

$$\Delta_{\mathrm{input}} = \begin{cases} \Delta_{\mathrm{output}} & |\mathrm{input}| < 1 \\ 0 & \mathrm{else} \end{cases} ,$$ (19)

where $\Delta_{\mathrm{output}}$ is gradient we get from the next layer and $\Delta_{\mathrm{input}}$ is the gradient we pass to the preceding layer. The conditions required for allowing the gradients information to propagate backward are discussed in appendix J. Those conditions are not enforced in this experiment, as they have no significant effect on the results, as shown in appendix H, where we add more results that isolate the forward-pass from the backward pass. Also included in appendix H are results that show the evolution of the training and test accuracy in training time. A simplified initialization scheme for the use of practitioners is included in appendix I.

We set the hidden layer width to 2048. We use SGD for training, a learning rate of $10^{-3}$ for networks with 10-90 layers, and a learning rate of $5 \times 10^{-4}$ when training 100-220 layers. Those parameters were selected to match those reported in [26], with the second learning rate adjusted to fit our area-of-search. We also use a batch size of 32, and use a standard preprocessing of the MNIST input[4].

Figure 3 shows that the initialization of the network using the parameters suggested by our theory achieves the optimal trainability when the number of layers is high. When measuring test accuracy at the early stage of the network, we can see that the accuracy is high when the network has $\sim 4\xi$ layers or less. As demonstrated by the advanced training stage results shown in appendix H, and by the results of [26], networks of depth exceeding $\sim 6\xi$ appear to be untrainable.

## 6    Discussion

In this paper, we study the effect of using quantized activations on the propagation of signals in deep neural networks, from the inputs to the outputs. We focus on quantized activations, which maps its input to a finite set of $N$ possible outputs. Our analysis suggests an initialization scheme that improves network trainability, and that fully-connected and convolutional networks to become untrainable when the number of layers exceeds $L_{\mathrm{max}} \sim 3\,(N+1)^{1.82}$.

Additionally, we propose a possible explanation for the improved generalization observed when training networks that are initialized to enable stable signal propagation. While the motivation for the critical initialization has been improved trainability [26], empirically these initialization schemes were shown to improve generalization as well, an observation that was beyond the scope of the analysis which motivated them. By considering the dynamics of learning in wide networks that

exhibit poor signal propagation, we find that generalization error in the early stages of training will typically not improve. This effect will be minimized when using a critical initialization.

The limitations of poor signal propagation can perhaps be overcome with certain modifications to the architecture or training procedure. Residual connections, for example, can be initialized [35] to maintain the signal propagation conditions even when the full-network depth exceeds our theoretical limit [31]. Another possible modification is batch normalization, which we did not consider in the analysis. While batch normalization by itself was shown to have negative side effects on the signal propagation [32], other studies [3, 6, 13] have already suggested that applying proper batch normalization is key when training quantized feed-forward networks. There are, however, cases where batch normalization does not work well, like in the case of recurrent neural networks. We expect our findings to have as increased significance if generalized to such architectures, as was done previously for continuous activations [4, 8].

### Acknowledgements

The work of DS was supported by the Israel Science foundation (grant No. 31/1031), the Taub Foundation and used a Titan Xp donated by the NVIDIA Corporation. The work of DG was supported by the NSF NeuroNex Award DBI-1707398 and the Gatsby Charitable Foundation. The work of DG and DS was done in part while the authors were visiting the Simons Institute for the Theory of Computing.

## Footnotes

[1]In principle the following results should hold under more generally mild moment conditions alone.

[2]When taking the sequential limit, asymptotic normality is a consequence of repeated application of the Central Limit Theorem [24]

[3]It will at times be convenient to consider the dynamics of the correlations of the *post-activations* $\hat{\alpha}^{(l)} = \phi(\alpha^{(l)})$ which we denote by $\widehat{\mathcal{M}}(\widehat{C})$. The rates of convergence are identical in both cases, as shown in Appendix B.

[4]The code for running this experiment and more is provided in `https://github.com/yanivbl6/quantized_meanfield`.

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
