[Supplementary Material]

# Appendix

## A  Proof of Lemma 1

*Proof of Lemma 1.* The dynamical system is given by

$$
\left( \begin{array}{c} Q^{(l)} \\ C^{(l)} \end{array} \right) = \left( \begin{array}{c} \sigma_w^2 \underset{u \sim \mathcal{N}(0,Q^{(l-1)})}{\mathbb{E}} \phi^2(u) + \sigma_b^2 \\ \frac{1}{Q^{(l-1)}} \left[ \sigma_w^2 \underset{(u_1,u_2) \sim \mathcal{N}(0,\Sigma(Q^{(l-1)},C^{(l-1)}))}{\mathbb{E}} \phi(u_1)\phi(u_2) + \sigma_b^2 \right] \end{array} \right) \equiv \mathcal{M} \left[ \left( \begin{array}{c} Q^{(l-1)} \\ C^{(l-1)} \end{array} \right) \right].
$$

(20)

Since $Q^{(l)} = \sigma_w^2 \widehat{Q}^{(l-1)} + \sigma_b^2$, convergence of $Q^{(l)}$ to a fixed point is equivalent to convergence of $\widehat{Q}^{(l)}$. If we assume $Q^{(l)}$ has converged to $Q^*$, the system in eq. 20 reduces to

$$
\mathcal{M}_{Q^*}(C) = \frac{1}{Q^*} \left[ \sigma_w^2 \underset{(u_1,u_2) \sim \mathcal{N}(0,\Sigma(Q^*,C))}{\mathbb{E}} \phi(u_1)\phi(u_2) + \sigma_b^2 \right]
$$

(21)

Linearizing the above equation gives

$$
\mathcal{M}_{Q^*}(C) = \mathcal{M}_{Q^*}(C^*) + \underbrace{\frac{\partial \mathcal{M}_{Q^*}(C^*)}{\partial C}}_{\equiv \chi}(C - C^*) + O\left((C - C^*)^2\right)
$$

and using a Cholesky decomposition and denoting by $\mathcal{D}x$ a standard Gaussian measure, we have

$$
\chi_{C^*} = \frac{1}{Q^*} \frac{\partial}{\partial C} \left[ \sigma_w^2 \underset{(u_1,u_2) \sim \mathcal{N}(0,\Sigma(Q^*,C))}{\mathbb{E}} \phi(u_1)\phi(u_2) + \sigma_b^2 \right]_{C=C^*}
$$

$$
= \frac{\sigma_w^2}{Q^*} \int \mathcal{D}z_1 \mathcal{D}z_2 \phi(\sqrt{Q_*}z_1 + \mu_b) \frac{\partial}{\partial C} \phi(\sqrt{Q_*}\left(Cz_1 + \sqrt{1-C^2}z_2\right) + \mu_b)_{C=C^*}
$$

$$
= \frac{\sigma_w^2}{Q^*} \int \mathcal{D}z_1 \mathcal{D}z_2 \phi(\sqrt{Q_*}z_1 + \mu_b) \phi'(\sqrt{Q_*}\left(Cz_1 + \sqrt{1-C^2}z_2\right) + \mu_b) \sqrt{Q_*} \left(z_1 - \frac{z_2 C}{\sqrt{1-C^2}}\right)
$$

and using $\int \mathcal{D}z g(z)z = \int \mathcal{D}z g'(z)$ which holds for any $g(z)$

$$
= \sigma_w^2 \underset{(u_1,u_2) \sim \mathcal{N}(0,\Sigma(Q^*,C))}{\mathbb{E}} \phi'(u_1)\phi'(u_2).
$$

The time scale of convergence dictated by the rate $\chi$ is obtained by solving the linear equation for $\varepsilon^{(l)} = C^{(l)} - C^*$, which gives $\varepsilon^{(l)} = \varepsilon_0 e^{-l/\xi}$ and thus in the linear regime we have

$$
e^{-1/\xi} = \frac{\varepsilon^{(l+1)}}{\varepsilon^{(l)}} = \frac{\mathcal{M}_{Q^*}(C^{(l)}) - C^*}{C^{(l)} - C^*} \approx \frac{C^* + \chi\left(C^{(l)} - C^*\right) - C^*}{C^{(l)} - C^*} = \chi
$$

$$
\xi = -\frac{1}{\log \chi}.
$$

Since a smooth convex function can intersect a linear function at no more than two points unless the two are equal (since otherwise the gradient must change sign twice implying negative curvature at some point), in order to show that $\mathcal{M}_{Q^*}(C)$ can have at most two fixed points in $[0,1]$ it suffices to show that it is convex in this range. A calculation similar to the one above gives:

$$
\frac{\partial^2 \mathcal{M}_{Q_*}(C)}{\partial C^2} = \sigma_w^2 Q_* \underset{(u_1,u_2) \sim \mathcal{N}(0,\Sigma(Q^*,C))}{\mathbb{E}} \phi''(u_1)\phi''(u_2).
$$

If $\phi$ is odd, so is $\phi''$ and then the expression above is non-negative for $C \in [0,1]$ according to Lemma 2 in [8]. It is obviously also non-negative simply if $\phi''$ is uniformly non-negative. The result applies to

quantized activation as well since we can replace the Heaviside function with a smooth approximation that is identical to within machine precision, and apply the above argument.

Since a fixed point is only stable if the slope $\chi$ is smaller than 1 and there are at most two fixed points in $[0, 1]$, there can be at most one stable fixed point. It follows that the fixed point of the dynamics does not depend on initialization as long as $C^{(0)} \geq 0$. While there may be another stable fixed point in $[-1, 0)$, the network will still be unable to distinguish between any two inputs that are either completely uncorrelated or positively correlated, which will generally prevent learning aside from trivial tasks where data points in different classes are always negatively correlated, and thus the data is linearly separable. $\qquad\square$

## B   Covariances of post-activations

In the main text we review results on asymptotic normality of pre-activations $\alpha^{(l)}(x)$ of deep feed-forward networks at the infinite width limit. The analysis of signal propagation in such networks is based on studying convergence of the covariances of these pre-activations to their fixed points. The convergence rate in eq. 6 and the corresponding time scale in eq. 7 that gives the typical maximal trainable depth are thus the main objects of interest.

It will be convenient at times to consider instead the evolution of the covariances of the post-activations $\widehat{\alpha}^{(l)}(x) = \phi(\alpha^{(l)}(x))$. We do this by defining, analogously to eq. 3,

$$\left( \begin{array}{cc} \mathbb{E}\widehat{\alpha}_i^{(l)}(x)\widehat{\alpha}_i^{(l)}(x) & \mathbb{E}\widehat{\alpha}_i^{(l)}(x)\widehat{\alpha}_i^{(l)}(x') \\ \mathbb{E}\widehat{\alpha}_i^{(l)}(x)\widehat{\alpha}_i^{(l)}(x') & \mathbb{E}\widehat{\alpha}_i^{(l)}(x')\widehat{\alpha}_i^{(l)}(x') \end{array} \right) = \left( \begin{array}{cc} \widehat{\Sigma}^{(l)}(x,x) & \widehat{\Sigma}^{(l)}(x,x') \\ \widehat{\Sigma}^{(l)}(x,x') & \widehat{\Sigma}^{(l)}(x',x') \end{array} \right) + \left(\widehat{\mu}^{(l)}\right)^2 \left( \begin{array}{cc} 1 & 1 \\ 1 & 1 \end{array} \right)$$

$$= \widehat{Q}^{(l)} \left( \begin{array}{cc} 1 & \widehat{C}^{(l)} \\ \widehat{C}^{(l)} & 1 \end{array} \right) + \left(\widehat{\mu}^{(l)}\right)^2 \left( \begin{array}{cc} 1 & 1 \\ 1 & 1 \end{array} \right) \tag{22}$$

For a given $x, x'$ the quantities $Q^{(l)}, C^{(l)}$ are trivially related to $\widehat{\mu}^{(l-1)}, \widehat{Q}^{(l-1)}, \widehat{C}^{(l-1)}$ via eq. 2, which gives

$$Q^{(l)} = \sigma_w^2 \left( \widehat{Q}^{(l-1)} + \left(\widehat{\mu}^{(l-1)}\right)^2 \right) + \sigma_b^2$$

$$C^{(l)} = \frac{\sigma_w^2 \left( \widehat{Q}^{(l-1)}\widehat{C}^{(l-1)} + \left(\widehat{\mu}^{(l-1)}\right)^2 \right) + \sigma_b^2}{Q^{(l)}}.$$

The covariance map for the hidden states analogous to eq. 5 is simply

$$\widehat{\mathcal{M}}_{\widehat{\mu}^*, \widehat{Q}^*}(\widehat{C}) = \frac{1}{\widehat{Q}^*} \mathop{\mathbb{E}}_{(u_1, u_2) \sim \mathcal{N}(\widehat{\mu}^*, \widehat{\Sigma}(\widehat{Q}^*, \widehat{C}))} \phi(u_1)\phi(u_2) \tag{23}$$

where $\widehat{\Sigma}(\widehat{Q}^*, \widehat{C}) = \left( \begin{array}{cc} \sigma_w^2\widehat{Q}^* + \sigma_b^2 & \sigma_w^2\widehat{Q}^*\widehat{C} + \sigma_b^2 \\ \sigma_w^2\widehat{Q}^*\widehat{C} + \sigma_b^2 & \sigma_w^2\widehat{Q}^* + \sigma_b^2 \end{array} \right)$. The convergence rates for 5 are identical since

$$\frac{\partial \mathcal{M}_{Q^*}(C^{(l)})}{\partial C^{(l)}} = \frac{\partial C^{(l+1)}}{\partial C^{(l)}} = \frac{1}{Q^*} \frac{\partial \sigma_w^2 \left( \widehat{Q}^*\widehat{C}^{(l)} \right) + \sigma_b^2}{\partial C^{(l)}}$$

$$= \frac{\sigma_w^2\widehat{Q}^*}{Q^*} \frac{\partial \widehat{C}^{(l)}}{\partial \widehat{C}^{(l-1)}} \frac{\partial \widehat{C}^{(l-1)}}{\partial C^{(l)}} = \frac{\partial \widehat{C}^{(l)}}{\partial \widehat{C}^{(l-1)}} = \frac{\partial \widehat{\mathcal{M}}_{\widehat{\mu}^*, \widehat{Q}^*}(\widehat{C}^{(l-1)})}{\partial \widehat{C}^{(l-1)}}$$

giving

$$\chi = \lim_{l \to \infty} \frac{\partial \mathcal{M}_{Q^*}(C^{(l)})}{\partial C^{(l)}} = \lim_{l \to \infty} \frac{\partial \widehat{\mathcal{M}}_{\widehat{\mu}^*, \widehat{Q}^*}(\widehat{C}^{(l-1)})}{\partial \widehat{C}^{(l-1)}} = \widehat{\chi}.$$

## C   Calculation of the fixed point slope for sign-activation

For convinience, we use the hidden states covariances and mapping $\widehat{C}, \widehat{Q}, \widehat{\mathcal{M}}$ as defined in appendix B, as they have a linear relationship to the pre-activation at the fixed point. Using a Cholesky decomposition on the equation 10: $\chi = 4\sigma_w^2 \underset{(u_a,u_b)\sim\mathcal{N}(0,\boldsymbol{\Sigma}(Q^*,C^*))}{\mathbb{E}} \delta(u_a)\delta(u_b)$, we get

$$4\sigma_w^2 \iint\limits_{u_1 \, u_2} \frac{1}{2\pi} \exp\left(-\frac{u_1^2 + u_2^2}{2}\right) \delta(\sqrt{Q^*}u_1)\delta\left(\sqrt{Q^*}\left(C^*u_1 + \sqrt{1-(C^*)^2}u_2\right)\right) du_1 du_2.$$

The delta functions enforces: $u_1 = 0, \mu_2 = 0$, giving us

$$\chi = \frac{2}{\pi} \frac{\sigma_w^2}{Q^*\sqrt{1-(C^*)^2}}.$$

Then, using $Q^* = \sigma_w^2 \widehat{Q}^* + \sigma_b^2$, and since $\widehat{Q}^* = 1$ for sign activation:

$$\chi = \frac{2}{\pi} \frac{\sigma_w^2}{(\sigma_w^2 + \sigma_b^2)\sqrt{1-(C^*)^2}}.$$

While this equation is written for the fixed point $C^*$, this equation can describe the slope of $\mathcal{M}(C)$ for every value of $C$. Rather than directly calculating $\mathcal{M}(C)$ using equation 4, it is surprisingly time saving to calculate it by using our expression for $\chi(C) = \frac{d\mathcal{M}(C)}{dC}$:

$$\mathcal{M}(C) - \text{const} = \int_0^C \chi(C')dC' = \frac{2}{\pi} \frac{\sigma_w^2}{(\sigma_w^2 + \sigma_b^2)} \arcsin(C).$$

We know that $\mathcal{M}(C = 1) = 1$, from which we can compute the constant

$$\text{const} = \mathcal{M}(1) - \frac{2}{\pi} \frac{\sigma_w^2}{(\sigma_w^2 + \sigma_b^2)} \arcsin(1) = \frac{\sigma_b^2}{\sigma_w^2 + \sigma_b^2}.$$

In conclusion:

$$\mathcal{M}(C) = \frac{\frac{2\sigma_w^2}{\pi} \arcsin(C) + \sigma_b^2}{\sigma_w^2 + \sigma_b^2}$$

It's also worth noting that for the hidden-states, the mapping for sign activation is:

$$\widehat{\mathcal{M}}(\widehat{C}) = \frac{2}{\pi} \arcsin\left(\frac{\widehat{C}\sigma_w^2 + \sigma_b^2}{\sigma_w^2 + \sigma_b^2}\right)$$

In addition to the fixed point $\widehat{\mathcal{M}}(\widehat{C} = 1) = 1$, the covariance mapping function suggests an additional fixed point within the range $[0, 1)$. In the case of $\sigma_b^2 = 0$, The entire network becomes anti-symmetric upon initialization and $C = -1$ becomes an infinitely unstable fixed point as well.

## D   Stochastic Rounding

One possible way to counter the negative effects of quantization which has proven itself in the past, is by adding noise to the rounding process. Being a commonplace method in machine learning, we would like to explore the effects of stochastic rounding on the dynamics of the neural network. When using this method the sign activation becomes probabilistic and can be modeled as:

$$\phi(x) = \text{sign}(x + n) \tag{24}$$

when $n \sim \text{Uniform}[-1, 1]$ is randomized for every neuron. Rather than working with a uniformly distributed noise, we replace it with a normal-distributed noise. Therefore, $\phi(u) = \text{sign}(u + n)$, for $n \sim \mathcal{N}(0, a^2)$. We justify this using a numeric simulation presented in figure 4, and in Appendix D.1

we find that the expression for stochastic rounding mapping (for hidden states) $\widehat{\mathcal{M}}_{sr}(\widehat{C})$ is:

$$\widehat{\mathcal{M}}_{sr}(\widehat{C}) = \frac{2}{\pi} \arcsin\left(\frac{1}{B} \frac{\widehat{C}\sigma_w^2 + \sigma_b^2}{\sigma_w^2 + \sigma_b^2}\right) \tag{25}$$

where $B = \sqrt{1 + \left(\frac{a}{Q^*}\right)^2 (2Q^* + a^2)}$. While the new mapping function for $\widehat{C}$ does not reach infinite slope at any point (since $C \leq 1, B > 1$), the noise also eliminates $\widehat{C} = 1$ as a fixed point. This result is consistent with the findings of [26] who have shown a similar phenomena when using dropout. Due to the arcsin function being a convex, monotonically increasing function in the area $0 < C < 1$, We can also conclude that adding noise (and therefore, increasing $B$) can only decrease the fixed point slope. See L.2 for proof, and figure 4 for illustration.

Figure 4: A simulation comparing $\widehat{\mathcal{M}}(\widehat{C})$ for deterministic and stochastic sign activations. For the Gaussian noise, we used the distribution $\mathcal{N}(0, \frac{1}{3})$, so both Gauss and Uniform stochastic activations have the same first and second moments. In all cases, the stochastic activation with the Gauss noise was indistinguishable from the one with the uniform noise.

### D.1 Development of the mean field equations for stochastic rounding

We now want to use the stochastic sign activation function to evaluate how it effects the $\widehat{\mathcal{M}}(C)$. Using equation 6, and we get:

$$\chi = 4\sigma_w^2 \int_{-\infty}^{\infty} dn_1 \int_{-\infty}^{\infty} dn_2 \int_{-\infty}^{\infty} du_1 \int_{-\infty}^{\infty} du_2 \frac{1}{2\pi} \frac{1}{2\pi a^2} \exp\left(-\frac{n_1^2}{2a^2}\right) \exp\left(-\frac{n_2^2}{2a^2}\right)$$

$$\exp\left(-\frac{u_1^2}{2}\right) \exp\left(-\frac{u_2^2}{2}\right) \delta\left(\sqrt{Q^*}u_1 + n_1\right) \delta\left(\sqrt{Q^*}\left(Cu_1 + \sqrt{1-(C)^2}u_2\right) + n_2\right)$$

We use the delta functions to enforce: $u_1 = -\frac{n_1}{\sqrt{Q^*}}, u_2 = -\frac{n_2 - C(n_1)}{\sqrt{Q^*}\sqrt{1-(C)^2}}$ and get:

$$\chi = \frac{2\sigma_w^2}{\pi Q^* a^2 \sqrt{1-(C)^2}} \frac{1}{2\pi} \int_{-\infty}^{\infty} \int_{-\infty}^{\infty} \exp\left(-\frac{n_1^2}{2a^2}\right) \exp\left(-\frac{n_2^2}{2a^2}\right) \tag{26}$$

$$\exp\left(-\frac{(n_1)^2}{2Q^*}\right) \exp\left(-\frac{(n_2 - C(n_1))^2}{2Q^*(1-(C)^2)}\right) dn_1 dn_2$$

Which can otherwise be written as:

$$\frac{2\sigma_w^2}{\pi Q^* a^2 \sqrt{1 - (C)^2}} \frac{1}{2\pi} \int_{-\infty}^{\infty} \int_{-\infty}^{\infty} \exp\left[-\frac{1}{2}D\right] dn_1 dn_2$$

$$D = \frac{n_1^2 \left(1 - (C)^2\right) \left(a^2 + (Q^*)^2\right) + n_2^2 Q^* \left(1 - (C)^2\right) + a^2 n_2^2 - 2a^2 n_1 n_2 C + a^2 n_1^2 (C)^2}{Q^* \left(1 - (C)^2\right) a^2}$$

So:

$$\chi = \frac{2\sigma_w^2}{\pi Q^* a^2 \sqrt{1 - (C)^2}} \frac{1}{2\pi} \int_{-\infty}^{\infty} \int_{-\infty}^{\infty} \exp\left[-\frac{1}{2} \frac{1}{Q^* \left(1 - (C)^2\right) a^2} \begin{pmatrix} n_1 & n_2 \end{pmatrix} \Sigma^{-1} \begin{pmatrix} n_1 \\ n_2 \end{pmatrix}\right] dn_1 dn_2 \tag{27}$$

$$\Sigma^{-1} = \begin{pmatrix} \left(1 - (C)^2\right) a^2 + \left(1 - (C)^2\right) (Q^*)^2 + a^2 (C)^2 & -a^2 C \\ -a^2 C & Q^* \left(1 - (C)^2\right) + a^2 \end{pmatrix}$$

Solving the Gaussian we get:

$$|\Sigma|^{-1} = |\Sigma^{-1}| = \frac{\left(Q^* \left(1 - (C)^2\right) + a^2\right)^2 - \left(a^2 C\right)^2}{\left(Q^* \left(1 - (C)^2\right) a^2\right)^2} = \tag{28}$$

$$\frac{(Q^*)^2 \left(1 - (C)^2\right)^2 + 2a^2 Q^* \left(1 - (C)^2\right) + a^4 - a^4 (C)^2}{\left(Q^* \left(1 - (C)^2\right) a^2\right)^2} =$$

$$\frac{(Q^*)^2 \left(1 - (C)^2\right)^2 + a^2 \left(2Q^* + a^2\right) \left(1 - (C)^2\right)}{(Q^*)^2 \left(1 - (C)^2\right)^2 a^4}$$

Resulting:

$$\chi = \frac{2\sigma_w^2}{\pi Q^* a^2 \sqrt{1 - (C)^2}} \frac{1}{2\pi} \left(2\pi \sqrt{|\Sigma|}\right) = \frac{2\sigma_w^2}{\pi Q^* a^2 \sqrt{1 - (C)^2}} \sqrt{\frac{(Q^*)^2 \left(1 - (C)^2\right) a^4}{(Q^*)^2 \left(1 - (C)^2\right) + a^2 \left(2Q^* + a^2\right)}} \tag{29}$$

And we finally get:

$$\chi = \frac{2\sigma_w^2}{\pi Q^* \sqrt{\left(1 - (C)^2\right) + \left(\frac{a}{Q^*}\right)^2 \left(2Q^* + a^2\right)}}$$

For the rest of this section, We will use the shortcut $B \equiv \sqrt{1 + \left(\frac{a}{Q^*}\right)^2 \left(2Q^* + a^2\right)}$ We can now write the equation as:

$$\chi = \frac{2\sigma_w^2}{\pi Q^* \sqrt{\left(1 - \left(\frac{C}{B}\right)^2\right)}} \tag{30}$$

for $x = \frac{C^*}{B}, \frac{dC}{dx} = B$

$$\widehat{\mathcal{M}}(\widehat{C}) = \int \frac{d\widehat{\mathcal{M}}(\widehat{C})}{d\widehat{C}} d\widehat{C} = \int \frac{d\widehat{\mathcal{M}}(C)}{dC} \frac{d\widehat{C}}{dC} dC = \int \chi \frac{d\widehat{C}}{dC} \frac{dC}{dx} dx$$

When we again drop the constant so $\widehat{\mathcal{M}}(\widehat{C} = 1) = 1$, and get:

$$\widehat{\mathcal{M}}(\widehat{C}) = \frac{2}{\pi} \arcsin\left(\frac{C}{B}\right) \tag{31}$$

Based on this equation, we can also use a Taylor expansion, to estimate $\widehat{C}^*$, and we get the solution:

$$\widehat{C}^* \simeq 1 - \frac{4}{\pi^2} \frac{\sigma_w^2}{Q^* B} \left( 1 + \sqrt{1 + \left(\frac{\pi}{2}\right)^4 (B^2 - B) \left(\frac{Q^*}{\sigma_w^2}\right)^2} \right) \tag{32}$$

# E   Calculations of $Q^{(l)}$ and $\chi$ for general quantized activations

We start by evaluating $\widehat{Q}$, the hidden-state covariance (see appendix B) for the general quantization activation function defined in 13, using equation 22

$$\widehat{Q}^{(l)} = \mathop{\mathbb{E}}_{u \sim \mathcal{N}(0, Q^{(l)})} \left( A + \sum_{i=1}^{N-1} H\left(u - g_i\right) h_i \right)^2 - \left(\mu^{(l)}\right)^2,$$

where:

$$\mu^{(l)} = \mathop{\mathbb{E}}_{u \sim \mathcal{N}(0, Q^{(l)})} \left( A + \sum_{i=1}^{N-1} H\left(u - g_i\right) h_i \right) = A + \sum_{i=1}^{N-1} h_i \Phi\left(-\frac{g_i}{\sqrt{Q^{(l)}}}\right). \tag{33}$$

Here, we use $\Phi$ as the normal cumulative distribution function. The constant $A$ cancels out, and we can expand the multiplication:

$$\widehat{Q}^{(l)} = \sum_{i=1}^{N-1} \sum_{j=1}^{N-1} h_i h_j \left( \mathbb{E}\left[ H\left(u - g_i\right) H\left(u - g_j\right) \right] - \Phi\left(-\frac{g_i}{\sqrt{Q^{(l)}}}\right) \Phi\left(-\frac{g_j}{\sqrt{Q^{(l)}}}\right) \right),$$

And since $H\left(u - g_i\right) H\left(u - g_j\right) = H\left(u - g_{\max(i,j)}\right)$

$$\widehat{Q}^{(l)} = \sum_{i=1}^{N-1} \sum_{j=1}^{N-1} h_i h_j \left( \Phi\left(-\frac{\max(g_i, g_j)}{\sqrt{Q^{(l)}}}\right) - \Phi\left(-\frac{g_i}{\sqrt{Q^{(l)}}}\right) \Phi\left(-\frac{g_j}{\sqrt{Q^{(l)}}}\right) \right). \tag{34}$$

$\Phi\left(-x\right) \Phi\left(-y\right) = \Phi\left(-\max\left(x, y\right)\right) \Phi\left(-\min(x, y)\right)$, so we can see that:

$$\Phi\left(-\max\left(x, y\right)\right) - \Phi\left(-x\right) \Phi\left(-y\right) = \Phi\left(-\max\left(x, y\right)\right) \left(1 - \Phi\left(-\min(x, y)\right)\right)$$

And by using the CDF property $\Phi(-x) = 1 - \Phi(x)$, we get

$$\widehat{Q}^{(l)} = \sum_{i=1}^{N-1} \sum_{j=1}^{N-1} h_i h_j \Phi\left(-\frac{\max(g_i, g_j)}{\sqrt{Q^{(l)}}}\right) \Phi\left(\frac{\min(g_i, g_j)}{\sqrt{Q^{(l)}}}\right), \tag{35}$$

from which we can easily compute $Q^{(l+1)}$. In Appendix F, we develop an approximation for $\mathcal{M}(\widehat{C})$. However, for our more immediate concerns, we will go straight to evaluating the equation for the fixed point slope, from eq. 6:

$$\chi = \sigma_w^2 \sum_{i=1}^{N-1} \sum_{j=1}^{N-1} \iint_{u_1, u_2 \sim \mathcal{N}(0, \Sigma(Q^*, C^*))} h_i h_j$$
$$\delta\left(\sqrt{Q^*} u_1 - g_i\right) \delta\left(\sqrt{Q^*}\left(C^* u_1 + \sqrt{1 - (C^*)^2} u_2\right) - g_j\right) =$$

$$\frac{\sigma_w^2}{2\pi \sqrt{Q^*}} \sum_{i=1}^{N-1} \sum_{j=1}^{N-1} \int \exp\left[-\frac{1}{2} \frac{g_i^2}{Q^*}\right] \exp\left[-\frac{1}{2} u_2^2\right] h_i h_j$$
$$\delta\left(\sqrt{Q^*}\left(C^* \frac{g_i}{\sqrt{Q^*}} + \sqrt{1 - (C^*)^2} u_2\right) - g_j\right) =$$

$$\frac{\sigma_w^2}{2\pi Q^*\sqrt{1-(C^*)^2}} \sum_{i=1}^{N-1}\sum_{j=1}^{N-1} \exp\left[-\frac{1}{2}\frac{g_i^2}{Q^*}\right]\exp\left[-\frac{1}{2}\frac{(g_j-C^*g_i)^2}{Q^*\left(1-(C^*)^2\right)}\right]h_ih_j$$

which can be simplified to:

$$\chi = \frac{\sigma_w^2}{2\pi Q^*\sqrt{1-(C^*)^2}}\sum_{i=1}^{N-1}\sum_{j=1}^{N-1} h_ih_j\exp\left[-\frac{g_i^2-2C^*g_ig_j+g_j^2}{2Q^*\left(1-(C^*)^2\right)}\right]. \tag{36}$$

## F  The general quantized activations mapping- Approximation and numeric evaluation

### F.1  The covariance mapping of a general quantized activation

We once again use the hidden states covariances $\widehat{Q},\widehat{C}$ Using eq. 5 for general quantized activation, we get the expression:

$$\widehat{C}^{(l)}\widehat{Q}^{(l)} = \underset{u_1,u_2\sim\mathcal{N}(0,\Sigma(Q^{(l)},C^{(l)}))}{\mathbb{E}}\left(\sum_{i=1}^{N-1}h_iH\left(u_1-g_i\right)-A\right)\left(\sum_{i=1}^{N-1}h_iH\left(u_2-g_j\right)-A\right)-\left(\mu^{(l)}\right)^2,$$

where we can use eq. 33 and expand it to:

$$\widehat{C}^{(l)}\widehat{Q}^{(l)} = \sum_{i=1}^{N-1}\sum_{j=1}^{N-1}\left(\underset{u_1,u_2\sim\mathcal{N}(0,\Sigma(Q^l,C^l))}{\mathbb{E}}\left[H\left(u_1-g_i\right)H\left(u_2-g_j\right)\right]-\Phi\left(\frac{-g_i}{\sqrt{Q^{(l)}}}\right)\Phi\left(\frac{-g_j}{\sqrt{Q^{(l)}}}\right)\right).$$

When the offsets are different than zero, there is no exact solution for the expectancy when $u_1,u_2$ are correlated. Article [29] suggests an approximation for finding $\widehat{\mathcal{M}}(\widehat{C})$, when $C(\widehat{C})=\frac{Q^*C\sigma_w^2+\sigma_b^2}{Q^*\sigma_w^2+\sigma_b^2}$:

$$\begin{aligned}\widehat{\mathcal{M}}(\widehat{C}) &\simeq \frac{\arcsin(C^*)}{2\pi Q^*}\sum_{i=1}^{N-1}\sum_{j=1}^{N-1}h_ih_j\cdot\\ &\exp\left(-\frac{1}{2}\frac{C}{\arcsin(C)Q^*\sqrt{1-C^2}}\left(g_i^2+g_j^2-g_ig_j\frac{2C}{1+\sqrt{1-C^2}}\right)\right)\end{aligned} \tag{37}$$

We found the approximation to hold well in the area $C\sim 0$, and $\forall i, g_i < Q^*$. Therefore, when $Q^*$ is known, this equation can be used to evaluate $C^*$ with reduced complexity.

### F.2  Quick numeric method to approximate the fixed point slope, for $\sigma_b > 0$

Using eq. 37, we suggest a numeric algorithm to evaluate the fixed point slope for $\sigma_b > 0$, for any quantized activation function:

1. Evaluate $Q$ by iterative usage of eq. 14. Start with arbitrary value $\widehat{Q}=1.0$ and repeat $T$ times.
2. Use eq. 37 to evaluate $\widehat{\mathcal{M}}(\widehat{C}=0)$ (Reminder: $C(\widehat{C}=0)=\frac{\sigma_b^2}{\widehat{Q}\sigma_w^2+\sigma_b^2}$)
3. Use eq. 15 to evaluate $\chi(\widehat{C}=0)$
4. Estimate $\widehat{C}^*$ by $\frac{C(\widehat{C}=0)}{1-\chi(\widehat{C}=0)}$ (First order approximation), and use equation 15 to find the fixed point slope.

We found this algorithm to be very efficient and accurate when studying the dynamics in the area of $\sigma_b > 0$. Results of using this estimation are displayed in figure 5.

## G  Beyond constant-spaced quantized activations

Our main focus in this article, have been the quantized activations with constant spacing. We now want to study the effects of using more complex activation functions on the dynamics of the network. We will do so by defining a new family of quantized activation functions, the linear-spacing

Figure 5: Grid-Approximation of the depth scale $\xi$ for constant-spaced activations of different quantization levels, as a function of the initialization parameters. $D = 1$ was used as the constant space between offsets. For this approximation, we used the algorithm described in F.2. It is apparent that the maximal depth scale for all quantization levels is achievable for $\sigma_b \simeq 0$.

activations- For any given values of $h, c_1 > 0, c_2 \in \mathbb{R}$, the function parameters in accordance with equation 13, are:

$$\forall i \in \{1, .., N-1\}, m \equiv \left(k - \frac{N}{2} - 2\right), h_i = h,$$
$$\tilde{g}_i = \tilde{D}_0 m \left(1 + \tilde{D}_1 |m|\right) \tag{38}$$

This family of functions can be thought of a second order generalization of the constant-spaced functions, which correspond to the special case $\tilde{D}_1 = 0$. This family of functions is important, as it also includes sigmoid-like quantized activation functions (given for values of $\tilde{D}_1 > 0$). To evaluate the dynamics of the new family, we again use eq. 16 and the depth scale definition eq. 7, and run a grid search over the normalized values of $\tilde{D}_1, \tilde{D}_0$, calculating the depth scale for each combination of parameters. The results of the grid search for several different quantization levels are presented in Figure 6. In all of the tested activations, the maximal depthscale that we found was identical, within numeric error range, to the maximal depthscale found for constant-spaced activations, indicating that the additional degree of freedom does not help improving the dynamical properties of the activation.

## H   Additional MNIST training-results

When studying the empirical effects of the initialization parameters on trainability when using a 10 states quantization, and seen that the longest trainable network is achieved when using the $\tilde{D}_{opt}$, the optimal normalized distance between offsets, as proposed by our theory. Additional test have been made to other quantization levels as well and gave similar results. It is unclear from the results, however, whether the degradation of deep networks is caused by the unoptimized propagation of the forward pass, or by the unoptimized backward pass. To isolate the effects of the forward pass which are of more interest to us, we measured the effects of $\sigma_w$ on a 10 states quantization once more, but

Figure 6: Evaluation of the depth scale $\xi$ for linear-spaced quantized activation, with the initialization $\sigma_b = 0$. The search resolution is $1000 \times 1000$ for each quantization level. The maximum depth-scale on each grid for square spacing activations is always achievable for the constant spacing as well, where $\tilde{D}_1 = 0$.

Figure 7: Test accuracy of a 10-states activation in feed-forward network, over the MNIST data-set, with different initialization values and optimized STE for backward propagation of gradients. When compared with the 3, we can see that adjusting the networks for better backward propagation of the gradients does not have a significant effect on the trainability of deep networks.

optimized the STE to allow clean gradient propagation using $\rho^{-1} = \sigma_w \sqrt{\text{erf}\left(\frac{1}{\sqrt{2Q^*}}\right)}$, when using $\sigma_w$ and $Q^*$ based on each run's initialization values. Figure 7 shows the results of this experiment, and confirms that the optimal initialization is dominated by the forward pass.

Figure 8: Time evolution of the test accuracy. Line 1&2: The evolution of the heat maps presented in figure 3, at an early stage of training (Training accuracy) . Bottom line: Test accuracy at an advanced stage of training (16000 steps), for the same deployment. Those results align with the results of [26], showing that even in a late stage of training, networks with layers exceeding $\sim 6\xi$ are untrainable.

# I Simplified Optimization of the initialization parameters

Sections 4 describes an algorithm for computation of the value of the initialization parameter $\sigma_w$, that would allow the best signal propagation in the network for any quantized activation function. However, when dealing with the constant spaced activation functions of the form:

$$\phi_N(x) = -1 + \sum_{i=1}^{N-1} \frac{2}{N-1} H\left(x - \frac{2}{N-1}\left(i - \frac{N}{2}\right)\right),$$

we find that our suggested method of initialization quickly converges to the *Xavier initialization* [9], as the quantization levels increases. For simple initialization, we suggest a small modification for the Xavier method that accounts for quantization: When $F_{in}$ and $F_{out}$ are the fan-in and fan-out of the layer, rather than simply computing the standard error for weights initialization using

$\sigma_w = \sqrt{\frac{2}{F_{in}+F_{out}}}$ as in the case of normal Xavier, we suggest that using a factor of

$$\alpha_N = 1 + \frac{1.23}{(N+0.2)^2}$$

(when $N$ is the number of activation states), so that:

$$\sigma_w = \alpha_N \cdot \sqrt{\frac{2}{F_{in} + F_{out}}}$$

We see that for the continuous case, our activation function becomes hard-tangent and our factor becomes $\lim_{N \to \infty} \alpha_N = 1$. $\alpha_N$ was estimated by computing the value $\sigma_w$ that ensures $\frac{D}{\sqrt{Q}} = \tilde{D}_{\text{opt}}$ for states ranging from 1 to 128, and fitting the results $\sigma_w(N)$ to the function $1 + \frac{a}{(N-b)^2}$, which behaved accordingly. For the case where the number of states is larger than 128, the factor $\alpha_N$ is small enough for the error to be irrelevant. Figure 10 shows a comparison between the standard Xavier and our modified initialization for 3-states activation, where $\alpha_N$ is at it's peak.

## J   Backwards signal propagation for straight through estimator

While we use quantized activations for the forward pass, the backward propagation of quantized neural networks is, in our case, done by straight through estimators (STE). When using constant-spaced quantized activations, we choose a STE to imitate the backward pass of the hard-tanh function:

$$\phi_\rho(x) = \begin{cases} -\rho^{-1} & x < -1 \\ x\rho & -1 \le x \le 1 \\ \rho^{-1} & x > 1 \end{cases}$$

where $\rho > 0$ is a parameter that controls the slope of the hard-tanh, so the backward equation is determined by the derivative:

$$\phi'_\rho(x) = \begin{cases} \rho & |x| < 1 \\ 0 & \text{else} \end{cases} \tag{39}$$

The moments of a random $N \times N$ matrix $\overline{A}$ are given by $m_{\mathbf{A}}^{(i)} = \frac{1}{N}\mathbb{E}\text{tr}\left(\overline{\mathbf{A}}^i\right)$. In the case of eq. 9, and our STE $\phi_\rho$, the equation is reduced to

$$m_{\mathbf{JJ}^T}^{(1)} = \frac{1}{N}\mathbb{E}\text{tr}\left(\phi'_\rho(\mathbf{u}^*)\mathbf{W}\left(\phi'_\rho(\mathbf{u}^*)\mathbf{W}\right)\right)$$

where $u_i^* \sim \mathcal{N}(0, Q^*)$ .i.d and $\mathbf{D}_{\phi'(\mathbf{u}^*)}$ is a diagonal matrix with $\phi'(\mathbf{u}^*)$ on the diagonal. This gives

$$m_{\mathbf{JJ}^T}^{(1)} = \sigma_w^2 \int \left(\phi'_\rho(\sqrt{Q^*}z)\right)^2 \mathcal{D}z$$

where $\mathcal{D}z = \frac{1}{\sqrt{2\pi}}\exp\left(\frac{-z^2}{2}\right)$. Then obtain:

$$m_{\mathbf{JJ}^T}^{(1)} = \sigma_w^2\rho^2 \int\limits_{-1/\sqrt{Q^*}}^{1/\sqrt{Q^*}} \mathcal{D}u = \sigma_w^2\rho^2\text{erf}\left(\frac{1}{\sqrt{2Q^*}}\right). \tag{40}$$

Assuming we already have the value $\sigma_w, Q^*$, we can set $\rho^{-1} = \sigma_w\sqrt{\text{erf}\left(\frac{1}{\sqrt{2Q^*}}\right)}$ to ensure $m_{\mathbf{JJ}^T} = 1$, and thus avoid vanishing and exploding gradients. In our main results, we avoided modifying the STE parameter $\rho$ in order to keep the experiment simple, and used the trivial STE using $\rho = 1$.

Figure 10: Comparison of our suggested initialization with the Xavier Gaussian initialization, for MNIST training using a 3-states quantized activation for layer numbers near the depth scale $6\xi_{max} \simeq 37$. For each number of layers and initialization, we used a grid search to find best learning rate from the values $[0.25, 0.5, 1, 2, 4, 10] \times 10^{-3}$, with all other run parameters as described in the experimental part of section 5. We ran 25 seeds using that learning rate, and the plot describes the mean and standard error of the test accuracy, at every step. In all cases, our suggested modification outperforms Xavier initialization by a small margin. With 40 layers, the network depth exceeds the theoretical depth scale, and all trainings fail under the 20000 steps limitation.

# K  Comparing convergence in $C$ and $Q$ directions

In previous papers studying signal propagation in feed-forward networks [24, 26, 30], it has been argued that the convergence in $Q$ direction is significantly faster than the convergence in the $C$ direction. Under this assumption, one can derive the approximate depth-scale by analyzing convergence in the $C$ direction only. The claim was established using empirical evidence [24] and using an approximated Taylor expansion of the activation function [26], by showing that the slope $\chi_c$ at $C^* = 1$ is always larger than the slope $\chi_q$ at $Q^*$. In our case, however, it is invalid to assume that the Taylor expansion of the quantized activation is correctly approximating the function behaviour, and either way $C^* = 1$ is an infinitely unstable fixed point and the convergence there can not be used as a baseline for comparison with the convergence in the $Q$ direction. It is therefore necessary to assert that this assumption holds for quantized activations as well. We will start by comparing $\chi_c, \chi_q$ analytically for general quantized activation function in the limit where the $\sigma_w$ is very small or very large, show that our assumption may fail in the case of some nontrivial activation functions and provide empirical evidence that the condition $\chi_c(C = C^*) > \chi_q(Q = Q^*)$ holds for trivial activation functions.

First, we argue that it is sufficient to show that $\chi_c(C = C^*) > \chi_q(Q = Q^*)$ for the depthscale in the $C$ direction ($\xi_C$) to be indicative of the full system-convergence. This is true because the mapping function of $Q$ is independent of the value of $C$. In the case of where $\chi_c(C = C^*) = \chi_q(Q = Q^*)$, we can, at the worst case, consider that $C$ will only start converging once $Q$ has converged, in which case the system would converge after a $K_c \xi_C + K_q \xi_q$ where $K_c, K_q$ are some constants.

Going back to eq. 15, using $\tilde{g}_i = \frac{g_i}{\sqrt{Q}}$, and picking the minimal value of $C = 0$ ($\mathcal{M}(C)$ is convex) :

$$\chi_c > \frac{\sigma_w^2}{Q} \sum_{i=1}^{N-1} \sum_{j=1}^{N-1} h_i h_j \frac{1}{2\pi} \exp\left[-\frac{\tilde{g_i}^2 + \tilde{g_j}^2}{2}\right] = $$
$$\frac{\sigma_w^2}{Q} \sum_{i=1}^{N-1} \sum_{j=1}^{N-1} h_i h_j \phi(\tilde{g}_i) \phi(\tilde{g}_j) \tag{41}$$

We do a similar derivation for the mapping of $Q$. From eq. 35, using $\frac{d\Phi(\frac{a}{\sqrt{x}})}{dx} = \frac{-a}{2x^{3/2}} \phi\left(\frac{a}{\sqrt{x}}\right)$, and denoting $G_{i,j}^+ = \max(\tilde{g}_i, \tilde{g}_j), G_{i,j}^- = min(\tilde{g}_i, \tilde{g}_j)$ we get that:

$$\chi_q = \frac{d\mathcal{M}(Q)}{dQ} = \frac{\sigma_w^2}{2Q} \sum_{i=1}^{N-1} \sum_{j=1}^{N-1} h_i h_j \left[G_{i,j}^+ \phi\left(G_{i,j}^+\right) \Phi\left(G_{i,j}^-\right) - G_{i,j}^- \phi\left(G_{i,j}^-\right) \Phi\left(-G_{i,j}^+\right)\right] \tag{42}$$

Combining those results, we get that:

$$\frac{\chi_q}{\chi_c} \leq \frac{1}{2} \frac{\sum_{i=1}^{N-1} \sum_{j=1}^{N-1} h_i h_j \left[G_{i,j}^+ \phi\left(G_{i,j}^+\right) \Phi\left(G_{i,j}^-\right) - G_{i,j}^- \phi\left(G_{i,j}^-\right) \Phi\left(-G_{i,j}^+\right)\right]}{\sum_{i=1}^{N-1} \sum_{j=1}^{N-1} h_i h_j \phi\left(G_{i,j}^+\right) \phi\left(G_{i,j}^-\right)} \tag{43}$$

From this result, we can immediately see that when taking $\sigma_w \to \infty$, resulting, $G_{i,j}^{-/+} \to 0$, we get that $\frac{\chi_q}{\chi_c} \to 0$, so $\chi_q \ll \chi_c$.

To analyze the behaviour of $\sigma_w \to 0$, we will consider the **continuous** activation functions:

$$\phi(x) = \begin{cases} x(1 - \alpha|x|) & |x| < A \\ 1 & \text{else} \end{cases} \tag{44}$$

where for $\alpha = 0$ we get an hard-tanh and for $\alpha > 0$ we get a sigmoid like function. The derivative of this function is:

$$\phi'(x) = \begin{cases} 1 - 2\alpha|x| & |x| < A \\ 0 & \text{else} \end{cases}. \tag{45}$$

We also calculate the derivative $\frac{\partial\mathcal{M}(Q)}{\partial Q}$ directly from eq. 4 and get:

$$\chi_q(Q) = \frac{1}{Q} \sigma_w^2 \mathbb{E}\left[\phi'(x) \phi(x) x\right] \tag{46}$$

where $X \sim \mathbb{N}(0, Q)$. We will also use the previous result $\chi_c(C) > \chi_c(0) = \sigma_w^2 \mathbb{E}\left[\phi'(x)^2\right]$. If we look at values where $\sigma_w$ is small, resulting small enough $Q$ so values outside the region $x < |A|$ can be ignored, and we get:

$$\chi_q = \frac{\sigma_w^2}{Q} \mathbb{E}\left[\phi'(x)\phi(x)x\right] = \frac{\sigma_w^2}{Q} \mathbb{E}\left[\left(1 - 3\alpha |x| + 2\alpha^2 x^2\right) x^2\right] \tag{47}$$

which sums up to:

$$\chi_q = \sigma_w^2 \left(1 - 6\alpha\sqrt{\frac{2Q}{\pi}} + 2\alpha^2 Q\,(3!!)\right) = \sigma_w^2 \left(1 - 6\alpha\sqrt{Q}\sqrt{\frac{2}{\pi}} + 6\alpha^2 Q\right) \tag{48}$$

Similarly,

$$\chi_c(C = 0) = \sigma_w^2 \mathbb{E}\left[\left(1 - 2\alpha E |x|\right)^2\right] = \sigma_w^2 \left(1 + 4\alpha^2 Q - 4\alpha\sqrt{Q}\sqrt{\frac{2}{\pi}}\right). \tag{49}$$

The condition $\chi_c \geq \chi_q$ therefore translates to:

$$1 + 4\alpha^2 Q - 4\alpha\sqrt{Q}\sqrt{\frac{2}{\pi}} > 1 - 6\alpha\sqrt{Q}\sqrt{\frac{2}{\pi}} + 6\alpha^2 Q \tag{50}$$

or simply $\alpha\sqrt{\frac{2}{\pi}} > \alpha^2\sqrt{Q}\left(3 - \frac{4}{\pi}\right)$. We can immediately see that in the non-trivial case of $\alpha < 0$, the activation functions will not comply with the condition ($Q$ can be infinitely small), and $Q$ may, indeed, converge slower than $C$. For $\alpha = 0$, we can see that the convergence of $Q$ and $C$ is identical. For the case of $\alpha > 0$, we get the new condition $\alpha\sqrt{Q} < \frac{\sqrt{\frac{2}{\pi}}}{\left(3 - \frac{4}{\pi}\right)} \simeq .462$

To see if this is true we need to estimate what is the region where our "small $Q$" assumption is valid. First, to keep the function continuous we can calculate $A = \frac{1 - \sqrt{1 - 4\alpha}}{2\alpha}$, and we will check the condition in the case $\sqrt{(Q)} = \frac{A}{3}$ (so the probability of $x > A$ is small), giving us the condition $.462 > \frac{1 - \sqrt{1 - 4\alpha}}{6} > \frac{1}{6}$ which is always true.

To conclude the analytical analysis, we saw that for large values of $Q$ (when $\sigma_w$ is large) $\chi_q > \chi_c$ for quantized activation functions, and that for small $\sigma_w, Q$ we can expect the convergence rates to match on trivial continuous activation functions. To check the intermediate range and to verify those results for quantized activation, we numerically calculate the values of $\chi_q, \chi_c$ using equations 41,42. Results of this experiment are shown in figure 12.

Figure 12: Empirical comparison of the convergence rate (Fixed point slope) of $C$ and $Q$ for different quantized activation function, and varying hyperparameters ($\sigma_w$). The activation function's offsets, for each value of $\beta$ ($\beta \propto \alpha$ from eq. 44) and for given number of activation states, is calculated by $g_i = \frac{2}{n-1} \left( i - \frac{n}{2} \right) \left( 1 + \frac{\beta}{n^2} \left| i - n/2 \right| \right)$. In accordance with our theoretical derivation, $\lim_{\sigma_w \to \infty} \frac{\chi_q}{\chi_c} = 0$ and $\lim_{\sigma_w \to 0} \frac{\chi_q}{\chi_c} \leq 1$ if $\alpha \geq 0$. Our results also show that the gap between $\chi_c$ and $\chi_q$ is generally wider when the number of activation states is low, and that the condition $\chi_c > \chi_q$ holds for all hyperparameters for all trivial activation functions ($\beta > 0$)

# L Additional Proofs

## L.1

*Proof that fixed point slope for sign activation can only be optimal for $\sigma_b = 0$.* We would like to prove the the optimal slope at the fixed point for sign activation can only be achieved when we take $\sigma_b$ to zero. First, we will use the implicit function theorem to calculate $\frac{d\widehat{C}^*}{d\sigma_b}$ ($\widehat{C}$ is the hidden states covariance, as described in appendix B), using the fixed point equation:

$$F(\widehat{C}^*, \sigma_b) = \widehat{C}^* - \frac{2}{\pi} \arcsin(C^*) = 0$$

when $C^* = \frac{\widehat{C}^* \sigma_w^2 + \sigma_b^2}{\sigma_w^2 + \sigma_b^2}, Q^* = \sigma_w^2 + \sigma_b^2$ :

$$\frac{\partial F}{\partial \widehat{C}^*} = 1 - \chi$$

When we $\chi$ can be expressed using 11. Also:

$$\frac{\partial F}{\partial \sigma_b} = -\frac{2}{\pi} \frac{1}{\sqrt{1 - (C^*)^2}} \frac{\partial C^*}{\partial \sigma_b} =$$

$$-\frac{2}{\pi} \frac{1}{\sqrt{1 - (C^*)^2}} \frac{2\sigma_b \sigma_w^2 \left(1 - \widehat{C}^*\right)}{(\sigma_w^2 + \sigma_b^2)^2} = -\chi \frac{2\sigma_b \left(1 - \widehat{C}^*\right)}{Q^*}$$

and using the implicit function theorem:

$$\frac{d\widehat{C}^*}{d\sigma_b} = -\frac{\frac{\partial F}{\partial \sigma_b}}{\frac{\partial F}{\partial \widehat{C}^*}} = \frac{\chi}{1 - \chi} \frac{2\sigma_b \left(1 - \widehat{C}^*\right)}{Q^*}$$

we can now use it to calculate:

$$\frac{d\chi}{d\sigma_b} = \frac{2\sigma_w^2}{\pi (\sigma_w^2 + \sigma_b^2) \sqrt{1 - (C^*)^2}} \left[ -\frac{2\sigma_b}{Q^*} + \frac{C^*}{1 - (C^*)^2} \frac{dC}{d\sigma_b} \right]$$

while:

$$\frac{dC}{d\sigma_b} = \frac{\sigma_w^2}{\sigma_w^2 + \sigma_b^2} \frac{dC^*}{d\sigma_b} + \frac{2\sigma_b}{\sigma_w^2 + \sigma_b^2} - \frac{C^* \sigma_w^2 + \sigma_b^2}{(\sigma_w^2 + \sigma_b^2)} 2\sigma_b =$$

$$\frac{1}{Q^*} \left( \sigma_w^2 \frac{dC^*}{d\sigma_b} + 2\sigma_b (1 - C^*) \right)$$

so:

$$\frac{d\chi}{d\sigma_b} = \frac{2\sigma_w^2}{\pi (Q^*)^2 \sqrt{1 - (C^*)^2}} \left[ +\frac{C^*}{1 - (C^*)^2} \left( \sigma_w^2 \frac{dC^*}{d\sigma_b} + 2\sigma_b (1 - C^*) \right) - 2\sigma_b \right] =$$

$$\frac{d\chi}{d\sigma_b} = \frac{2\sigma_w^2}{\pi\,(Q^*)^2\,\sqrt{1-(C^*)^2}}\left[+\frac{C^*}{1-(C^*)^2}\left(\sigma_w^2\frac{\chi 2\sigma_b}{1-\chi}\left(1-C^*\right)\right)+2\sigma_b\frac{C^*}{1+(C^*)}-2\sigma_b\right] =$$

$$\frac{2\sigma_w^2}{\pi\,(Q^*)^2\,(1+C^*)\,\sqrt{1-(C^*)^2}}\left[\frac{C^*}{1-C^*}\sigma_w^2\frac{\chi}{1-\chi}\frac{2\sigma_b\left(1-\widehat{C}^*\right)}{Q^*}-2\sigma_b\right] =$$

$$\frac{4\sigma_w^2\sigma_b}{\pi\,(Q^*)^2\,(1+C^*)\,\sqrt{1-(C^*)^2}}\left[\frac{C^*Q^*}{\left(1-\widehat{C}^*\right)\sigma_w^2}\sigma_w^2\frac{\chi}{1-\chi}\frac{\left(1-\widehat{C}^*\right)}{Q^*}-1\right] =$$

$$\frac{2\sigma_b\chi}{(Q^*)\,(1+C^*)}\left[\frac{\chi C^*}{1-\chi}-1\right]$$

we learn that $sign\left(\frac{d\chi}{d\sigma_b}\right)$ depends on $\frac{\chi C^*}{1-\chi}-1$. if for some value of $\sigma_b, \sigma_w$, $\frac{\chi C^*}{1-\chi}>1$, then, $\frac{\chi C^*}{1-\chi}-1$ will remain positive when increasing $\sigma_b$, since $\frac{d\chi}{d\sigma_b}>0$ and $\frac{dC^*}{d\sigma_b}>0$ results $\frac{d}{d\sigma_b}\frac{\chi C^*}{1-\chi}>0$. The optimal (highest) value of $\chi$ for the given value of $\sigma_w$ will therefore be achieved in the limit $\sigma_b \to \infty$, and we can use the slope equation to calculate it:

$$\lim_{\sigma_b\to\infty}\chi = \lim_{\sigma_b\to\infty}\frac{2\sigma_w^2}{\pi\sqrt{\left(\sigma_w^2+\sigma_b^2\right)^2-\left(\widehat{C}^*\sigma_w^2+\sigma_b^2\right)^2}} = 0$$

(for this we use the fact that $\widehat{C}^* > 0$ for $\sigma_b > 0$)

And this contradicts our assumption that this is the highest value of $\chi$, so $\frac{d\chi}{d\sigma_b}$ must be negative for all values of $\sigma_b, \sigma_w$.

$\square$

## L.2

*Proof that stochastic rounding results smaller slope at the fixed point.* We have shown that the the covariance mapping function with stochastic rounding is $\widehat{\mathcal{M}}(\widehat{C}) = f(\frac{C_u}{B})$, when we denote $C_u(\widehat{C}) = \frac{\widehat{C}\sigma_w^2+\sigma_b^2}{\sigma_w^2+\sigma_b^2}$, $C_u^* = C_u(\widehat{C}^*)$, and $f(x)$ is a convex function for $0 \le x \le 1$ and the variable $B \ge 1$ is increasing as the variance of the stochastic rounding increase, and $B = 1$ gives us the mapping for deterministic function. We will show that $\frac{d\chi^*}{dB} < 0$, when $\chi^*$ is the fixed point slope. Using the implicit function theorem as we did in proof L.1, for the function:

$$F(\widehat{C}^*, B) = \widehat{C}^* - \widehat{\mathcal{M}}(\widehat{C}^*) = 0$$

for $\frac{\partial F}{\partial \widehat{C}^*}$ we get:

$$\frac{\partial F}{\partial \widehat{C}^*} = 1 - \chi^* > 0$$

when we used the definition of $\chi^*$ as the fixed point slop. For $\frac{\partial F}{\partial B}$, we get

$$\frac{\partial F}{\partial B} = -f'\left(\frac{C_u}{B}\right)\cdot\frac{-C_u}{B^2} > 0$$

using the implicit function theorem:

$$\frac{d\widehat{C}^*}{dB} = -\frac{\frac{dF}{dB}}{\frac{dF}{d\widehat{C}^*}} < 0$$

and since $\frac{d\widehat{C}_u^*}{d\widehat{C}^*} = \frac{\sigma_w^2}{\sigma_w^2 + \sigma_b^2} > 0$ this also means that:

$$\frac{dC_i^*}{dB} < 0 \tag{51}$$

from eq. 30 we know that:

$$\chi^* = \frac{2\sigma_w^2}{\pi Q^* \sqrt{\left(1 - \left(\frac{C_u^*}{B}\right)^2\right)}}$$

we can immediately see that for $\overline{C} \equiv \frac{C_u^*}{B}$, we get $\frac{d\chi^*}{d\overline{C}} > 0$, and from eq. 51 we get that $\frac{d\overline{C}}{dB} = \frac{B\frac{dC^*}{dB} - C_u^*}{B^2} < 0$ so the chain rule gives us $\frac{d\chi^*}{dB} < 0$. $\qquad\square$

# M  Neural tangent kernel for quantized activations

We consider the dynamics of training for deep, wide neural networks. We argue that the error at an average test point will not improve during early stages of training if the signal propagation conditions are not satisfied, and thus ensuring signal propagation should have a beneficial effect on generalization error.

## M.1  NTK setup

We consider full-batch gradient descent with regression loss in a continuous time setting. Defining a fitting error $\zeta_i = f(x_i) - y_i$ [5], the loss function is given by

$$\varphi = \frac{1}{2N_d} \sum_{i=1}^{N_d} \zeta_i^2.$$

where $N_d$ is the number of data points. The weights evolve in time according to

$$\frac{\partial \theta_p}{\partial t} = -\frac{\partial \varphi}{\partial \theta_p} = -\frac{1}{N_d} \sum_{i=1}^{N_d} \frac{\partial f(x_i)}{\partial \theta_p} \zeta_i$$

for all weights $\theta_p$. The evolution of the network function is then given by

$$\frac{\partial f(x_i)}{\partial t} = \sum_p \frac{\partial f(x_i)}{\partial \theta_p} \frac{\partial \theta_p}{\partial t} = -\frac{1}{N_d} \sum_p \frac{\partial f(x_i)}{\partial \theta_p} \frac{\partial f(x_j)}{\partial \theta_p} \zeta_i \equiv -\frac{1}{N_d} [\Theta \zeta]_i$$

where $p$ indexes all the weights of the neural network and we have defined the Gram matrix $\Theta \in \mathbb{R}^{N_d \times N_d}$ by

$$\Theta(x_i, x_j) = \sum_p \frac{\partial f(x_i)}{\partial \theta_p} \frac{\partial f(x_j)}{\partial \theta_p}. \tag{52}$$

This matrix is referred to as the Neural Tangent Kernel (NTK) in [14]. When considering this object at the infinite width limit, it is convenient to adopt the following parametrization for a fully connected network $f : \mathbb{R}^{n_0} \to \mathbb{R}^{n_{L+1}}$:

$$\phi(\alpha^{(0)}(x)) = x$$
$$\alpha^{(l)}(x) = \frac{\sigma_w}{\sqrt{n_{l-1}}} W^{(l)} \phi(\alpha^{(l-1)}(x)) + \sigma_b b^{(l)}, \quad l = 1, ..., L \tag{53}$$
$$f(x) = \alpha^{(L+1)}(x)$$

for input $x \in \mathbb{R}^{n_0}$ and weight matrices $W^{(l)} \in \mathbb{R}^{n_l \times n_{l-1}}$. The weights are initialized using $W_{ij}^{(l)} \sim \mathcal{N}(0,1), b_i^{(l)} \sim \mathcal{N}(0,1)$. The output of this NTK network is identical to that of a standard network, yet the gradients are rescaled such that $\Theta$ remains finite when taking the infinite width limit. For an appropriately chosen learning rate the dynamics of learning in the NTK network can be made identical to those of a standard network [16].

In [14], under some technical conditions, $\Theta$ was shown to be essentially constant during training at the sequential limit $\lim_{n_{L-1} \to \infty} \ldots \lim_{n_2 \to \infty} \lim_{n_1 \to \infty}$. At this limit, adapting Theorem 1 of [14] to allowing arbitrary variances for the weights and biases, one obtains the following asymptotic form of $\Theta$ at the sequential infinite width limit:

$$\overline{\Theta}(x, x') = \sum_{l=1}^{L+1} \overset{L+1}{\underset{j=l+1}{\Pi}} \Sigma'^{(j)}(x, x') \Sigma^{(l)}(x, x') \tag{54}$$

where

$$
\begin{aligned}
\Sigma^{(1)}(x, x') &= \frac{\sigma_w^2}{n_0} x^T x' + \sigma_b^2 \\
\Sigma^{(l)}(x, x') &= \sigma_w^2 \underset{(u_1, u_2) \sim \mathcal{N}(0, \Sigma^{(l-1)}|_{x,x'})}{\mathbb{E}} \phi(u_1)\phi(u_2) + \sigma_b^2 \\
\Sigma^{(l)}\big|_{x,x'} &= \left( \begin{array}{cc} \Sigma^{(l)}(x, x) & \Sigma^{(l)}(x, x') \\ \Sigma^{(l)}(x, x') & \Sigma^{(l)}(x', x') \end{array} \right).
\end{aligned}
\tag{55}
$$

are the covariances of the pre-activations and

$$\Sigma'^{(l)}(x, x') = \sigma_w^2 \underset{(u_1, u_2) \sim \mathcal{N}(0, \Sigma^{(l)}|_{x,x'})}{\mathbb{E}} \phi'(u_1)\phi'(u_2).$$

In [2] it was also shown that for finite width ReLU networks $\mathbb{E}\Theta = \overline{\Theta}$ and concentrates about its expectation with the fluctuations scaling inversely with layer width. It follows that when taking the layer widths to infinity in arbitrary order for ReLU networks one recovers $\overline{\Theta}$, and empirically $\Theta$ concentrates well around $\overline{\Theta}$ for other choices of nonlinearities [16]. We note that even when using the standard scaling 1, for very wide networks where the effect of individual weights will be negligible, even though the asymptotic for of the NTK at infinite width may be different, it will still change little in the initial phases of training.

## M.2  Continuous activations

We write the NTK for a feed-forward network in the NTK parametrization 53, omitting the dependence on $x$ of $f, \alpha^{(l)}$ to lighten notation

$$\frac{\partial f}{\partial W_{ij}^{(l)}} = \frac{\sigma_w}{\sqrt{n_{l-1}}} \frac{\partial f}{\partial \alpha_i^{(l)}} \phi(\alpha_j^{(l-1)}) = \frac{\sigma_w}{\sqrt{n_{l-1}}} \frac{\partial f}{\partial \phi(\alpha_i^{(l)})} \frac{\partial \phi(\alpha_i^{(l)})}{\partial \alpha_i^{(l)}} \phi(\alpha_j^{(l-1)})$$

$$= \sum_{k=1}^{n_{l+1}} \frac{\sigma_w}{\sqrt{n_{l-1}}} \frac{\partial f}{\partial \alpha_k^{(l+1)}} \frac{\partial \alpha_k^{(l+1)}}{\partial \phi(\alpha_i^{(l)})} \frac{\partial \phi(\alpha_i^{(l)})}{\partial \alpha_i^{(l)}} \phi(\alpha_j^{(l-1)})$$

$$= \frac{\sigma_w}{\sqrt{n_{l-1}}} \sum_{k=1}^{n_{l+1}} \frac{\partial f}{\partial \alpha_k^{(l+1)}} \frac{\sigma_w}{\sqrt{n_l}} W_{ki}^{(l+1)} \frac{\partial \phi(\alpha_i^{(l)})}{\partial \alpha_i^{(l)}} \phi(\alpha_j^{(l-1)})$$

restoring the $x$ dependence and defining a diagonal matrix $D^{(l)}(x) = \mathrm{diag}(\frac{\sigma_w}{\sqrt{n_l}} \frac{\partial \phi(\alpha_i^{(l)}(x))}{\partial \alpha_i^{(l)}(x)})$ we have

$$\frac{\partial f(x)}{\partial W_{ij}^{(l)}} = \frac{\sigma_w}{\sqrt{n_{l-1}}} \left[ \left( \frac{\partial f(x)}{\partial \alpha^{(l+1)}(x)} \right)^T W^{(l+1)} D^{(l)}(x) \right]_i \phi(\alpha_j^{(l-1)}(x))$$

we can repeat the process for the elements of $\frac{\partial f(x)}{\partial \alpha^{(l+1)}}$ finally obtaining

$$\frac{\partial f(x)}{\partial W_{ij}^{(l)}} = \frac{\sigma_w}{\sqrt{n_{l-1}}} \left[ W^{(L+1)} D^{(L)}(x) W^{(L)} ... W^{(l+1)} D^{(l)}(x) \right]_i \phi(\alpha_j^{(l-1)}(x)) \equiv \frac{\sigma_w}{\sqrt{n_{l-1}}} \widehat{\beta}_i^{(l)}(x) \widehat{\alpha}_j^{(l)}(x)$$

and we similarly obtain

$$\frac{\partial f(x)}{\partial b_i^{(l)}} = \sigma_b \widehat{\beta}_i^{(l)}(x).$$

The NTK thus takes the form

$$\Theta(x, x') = \sum_{l,i_l} \frac{\partial f(x)}{\partial W_{i_l i_{l-1}}^{(l)}} \frac{\partial f(x')}{\partial W_{i_l i_{l-1}}^{(l)}} + \sum_{l,i_l} \frac{\partial f(x)}{\partial b_{i_l}^{(l)}} \frac{\partial f(x')}{\partial b_{i_l}^{(l)}}$$

$$= \sigma_w^2 \sum_{l=1}^{L+1} \frac{1}{n_{l-1}} \left\langle \widehat{\beta}^{(l)}(x), \widehat{\beta}^{(l)}(x') \right\rangle \left\langle \widehat{\alpha}^{(l)}(x), \widehat{\alpha}^{(l)}(x') \right\rangle + \sigma_b^2 \left\langle \widehat{\beta}^{(l)}(x), \widehat{\beta}^{(l)}(x') \right\rangle$$

According to [14, 2], this tends to 54 at the infinite width limit.

## M.3  Quantized activations

We now consider dynamics in function space with quantized activations. Analyzing a single network in this fashion is hopeless since the network function is not a continuous function of the weights and so the dynamics will not be continuous. We can instead consider a stochastic rounding scheme where the post-activations are defined according to

$$\widehat{\alpha}_i^{(l)} = \text{sign}(\alpha_i^{(l)} - z_i^{(l)})$$

and $z_i^{(l)} \sim \text{Unif}([-1, 1])$. The connection between this setup and the straight-through estimator (STE) was first observed in [13]. We denote the set of all $z_i^{(l)}$ by $\{z\}$. Considering the dynamics of an ensemble average such that the loss function is given by

$$\varphi = \frac{1}{2N_d} \sum_{i=1}^{N_d} (\mathbb{E}_{\{z\}} f(x_i) - y_i)^2 = \frac{1}{2N_d} \sum_{i=1}^{N_d} \zeta_i^2$$

We have

$$\frac{\partial \mathbb{E}_{z_i^{(l)}} f}{\partial \alpha_i^{(l)}} = \frac{\partial}{\partial \alpha_i^{(l)}} \left( p(\alpha_i^{(l)} - z_i^{(l)} > 0 | \alpha_i^{(l)}) \left. f \right|_{\widehat{\alpha}_i^{(l)} = 1} + (1 - p(\alpha_i^{(l)} - z_i^{(l)} > 0 | \alpha_i^{(l)})) \left. f \right|_{\widehat{\alpha}_i^{(l)} = -1} \right)$$

$$= \frac{\partial p(\alpha_i^{(l)} - z_i^{(l)} > 0 | \alpha_i^{(l)})}{\partial \alpha_i^{(l)}} \left( \left. f \right|_{\widehat{\alpha}_i^{(l)} = 1} - \left. f \right|_{\widehat{\alpha}_i^{(l)} = -1} \right) = \frac{1}{2} \mathbb{1}_{|\alpha_i^{(l)}| \leq 1} \left( \left. f \right|_{\widehat{\alpha}_i^{(l)} = 1} - \left. f \right|_{\widehat{\alpha}_i^{(l)} = -1} \right).$$

If we now consider any smooth extension of $\gamma$ of $\widehat{\alpha}_i^{(l)}$ such that $[-1, 1] \subseteq \text{Im}(\gamma)$ and denote by $\tilde{f}$ a copy of $f$ where we replace $\widehat{\alpha}_i^{(l)}$ by $\gamma$. We then have

$$\left. f \right|_{\widehat{\alpha}_i^{(l)} = 1} - \left. f \right|_{\widehat{\alpha}_i^{(l)} = -1} = \left. \tilde{f} \right|_{\gamma=1} - \left. \tilde{f} \right|_{\gamma=-1} = \left. \frac{\partial \tilde{f}}{\partial \gamma} \right|_{\gamma=0} + \mathcal{O}\left( \left. \frac{\partial^3 \tilde{f}}{\partial \gamma^3} \right|_{\gamma=0} \right) = \left. \frac{\partial \tilde{f}}{\partial \gamma} \right|_{\gamma=0} + \mathcal{O}\left( \left. \frac{\partial^3 \tilde{f}}{\partial \gamma^3} \right|_{\gamma=0} \right)$$

$$= \left.\frac{\partial\tilde{f}}{\partial\gamma}\right|_{\gamma=\pm1} + \mathcal{O}\left(\left.\frac{\partial^2\tilde{f}}{\partial\gamma^2}\right|_{\gamma=0}\right) = \left.\frac{\partial f}{\partial\widehat{\alpha}_i^{(l)}}\right|_{\widehat{\alpha}_i^{(l)}=\pm1} + \mathcal{O}\left(\left.\frac{\partial^2\tilde{f}}{\partial\gamma^2}\right|_{\gamma=0}\right) = \frac{\partial\mathbb{E}_{z_i^{(l)}}f}{\partial\widehat{\alpha}_i^{(l)}} + \mathcal{O}\left(\left.\frac{\partial^2\tilde{f}}{\partial\gamma^2}\right|_{\gamma=0}\right).$$

If we neglect these higher order terms (which should be small since the influence of a single neuron on the output is generally small, and should vanish at the infinite width limit) and note that the above approximation holds if we condition on $\{z\}\backslash\{z_i^{(l)}\}$, we obtain

$$\frac{\partial\mathbb{E}_{\{z\}}f}{\partial\alpha_i^{(l)}} \approx \mathbb{1}_{\left|\alpha_i^{(l)}\right|\leq1}\frac{\partial\mathbb{E}_{\{z\}}f}{\partial\widehat{\alpha}_i^{(l)}}. \tag{56}$$

We can now repeat the calculation of the NTK using eq. 56, obtaining

$$\frac{\partial\mathbb{E}_{\{z\}}f}{\partial W_{ij}^{(l)}} = \frac{\sigma_w}{\sqrt{n_{l-1}}}\frac{\partial\mathbb{E}_{\{z\}}f}{\partial\alpha_i^{(l)}}\phi(\alpha_j^{(l-1)}) \approx \frac{\sigma_w}{\sqrt{n_{l-1}}}\frac{\partial f}{\partial\phi(\alpha_i^{(l)})}\mathbb{1}_{\left|\alpha_i^{(l)}\right|\leq1}\phi(\alpha_j^{(l-1)})$$

$$= \frac{\sigma_w}{\sqrt{n_{l-1}}}\sum_{k=1}^{n_{l+1}}\frac{\partial f}{\partial\alpha_k^{(l+1)}}\frac{\sigma_w}{\sqrt{n_l}}W_{ki}^{(l+1)}\mathbb{1}_{\left|\alpha_i^{(l)}\right|\leq1}\phi(\alpha_j^{(l-1)}).$$

Defining $D_{\text{STE}}^{(l)}(x) = \text{diag}(\frac{\sigma_w}{\sqrt{n_l}}\mathbb{1}_{\left|\alpha_i^{(l)}\right|\leq1})$ and applying eq. 56 repeatedly at each layer up until $L+1$ gives

$$\frac{\partial f(x)}{\partial W_{ij}^{(l)}} \approx \frac{\sigma_w}{\sqrt{n_{l-1}}}\left[W^{(L+1)}D_{\text{STE}}^{(L)}(x)W^{(L)}...W^{(l+1)}D_{\text{STE}}^{(l)}(x)\right]_i\phi(\alpha_j^{(l-1)}(x)) \equiv \frac{\sigma_w}{\sqrt{n_{l-1}}}\widehat{\beta}_{\text{STE},i}^{(l)}(x)\widehat{\alpha}_j^{(l)}(x)$$

$$\frac{\partial f(x)}{\partial b_i^{(l)}} \approx \sigma_b\widehat{\beta}_{\text{STE},i}^{(l)}(x).$$

and thus applying 52 gives

$$\frac{\partial\mathbb{E}_{\{z\}}f(x)}{\partial t} \approx -\frac{1}{N_d}\sum_i\Theta_{\text{STE}}(x,x_i)\zeta_i$$

where

$$\Theta_{\text{STE}}(x,x') = \sigma_w^2\sum_{l=1}^{L+1}\frac{1}{n_{l-1}}\left\langle\widehat{\beta}_{\text{STE}}^{(l)}(x),\widehat{\beta}_{\text{STE}}^{(l)}(x')\right\rangle\left\langle\widehat{\alpha}^{(l)}(x),\widehat{\alpha}^{(l)}(x')\right\rangle + \sigma_b^2\left\langle\widehat{\beta}_{\text{STE}}^{(l)}(x),\widehat{\beta}_{\text{STE}}^{(l)}(x')\right\rangle.$$

A trivial generalization of the calculation of the asymptotic form of $\Theta(x,x')$ at the infinite width limit in [14] shows that at this limit $\Theta_{\text{STE}}(x,x')$ tends to

$$\overline{\Theta}_{\text{STE}}(x,x') = \sum_{l=1}^{L+1}\prod_{j=l+1}^{L+1}\Sigma_{\text{STE}}^{'(j)}(x,x')\Sigma^{(l)}(x,x') \tag{57}$$

where $\Sigma^{(l)}(x,x')$ is defined in eq. 55,

$$\Sigma_{\text{STE}}^{'(l)}(x,x') = \sigma_w^2\mathop{\mathbb{E}}_{(u_1,u_2)\sim\mathcal{N}(0,\Sigma^{(l)}|_{x,x'})}\phi_{\text{STE}}'(u_1)\phi_{\text{STE}}'(u_2).$$

and we define the hard-tanh function,

$$\phi_{\text{STE}}(x) = \begin{cases} 1 & 1 \leq x \\ x & -1 < x < 1 \\ -1 & x \leq -1 \end{cases}.$$  (58)

for which $\phi'_{\text{STE}}(y) = \mathbb{1}_{|y| \leq 1}$. The form of $\overline{\Theta}_{\text{STE}}(x, x')$ is thus obtained by replacing the sign activation with eq. 58 but only during the backwards pass (and not during the forward pass), in line with the motivation of the STE in [13]. We note that the dynamics of this ensemble average correspond to those of the update scheme in eq. 19 with $\rho = 1$. Other choices will introduce a dependence on $\rho$ in $\overline{\Theta}_{\text{STE}}(x, x')$ but will not change the fact that it can be expressed as a function of the covariances of the inputs in eq. 55.

### M.4 Asymptotic NTK and generalization

We now consider a very deep network such that the covariance map approaches its fixed point

$$\Sigma^*|_{x,x'} = Q^* \begin{pmatrix} 1 & C^* \\ C^* & 1 \end{pmatrix}.$$

$\Theta^{(l)}(x, x')$ for very deep networks will approach a matrix of the form

$$\lim_{L \to \infty} \frac{1}{L+1} \Theta^{(L+1)}(x, x') = \Theta^*(x, x') = \alpha \delta(x, x') + \beta(1 - \delta(x, x'))$$  (59)

for some constants $\alpha, \beta$ and $\delta(x, x')$ is a Kronecker delta.

To understand the generalization properties of such a network, we can consider the evolution of the error at some test point $z$ that is not part of the training set. It will be given by

$$\frac{\partial \zeta(z)}{\partial t} = -\frac{L}{N_d} \sum_{i=1}^{N_d} \Theta^*(z, x_i) \zeta(x_i) = -\frac{\beta L}{N_d} \sum_{i=1}^{N_d} \zeta(x_i)$$

which at initialization is independent of our choice of $z$. Since it is also independent of the true label of $z$ this will mean that the generalization error will typically not decrease [6].

We conclude that for networks deep enough that the covariance map converges, in the initial phase of training before $\Theta$ changes considerably there will be no improvement in the generalization error at a typical test point. Conversely, this suggests that satisfying the signal propagation condition $\chi = 1$ will facilitate generalization. Presumably, if convergence to the fixed point is slow, instead of the form in eq. 59, $\Theta$ will exhibit some finite scale of decay from its value on the diagonal as a function of the distance between the inputs. This will enable points in the training set near $z$ that share the same label, and where the error has the same sign as $\zeta(z)$, to influence $\frac{\partial \zeta(z)}{\partial t}$ thus reducing the error at $z$. This argument is independent of the value of $\beta$, and provides further motivation for the study of critical initialization schemes that exhibit slow convergence to the fixed point [26]. Such initialization schemes have also been motivated in the past by concerns of trainability (i.e. ensuring stable signal propagation from the inputs to the hidden states of a deep network, and preventing vanishing/exploding gradients). This phenomenon could perhaps be the basis for the improvements in generalization observed when using critical initialization schemes, which have hitherto been unexplained.

To explore whether rapid convergence of the covariance map is correlated with a lack of structure in the NTK, we define a coarse metric for non-trivial structure in the off-diagonal terms of the NTK that should facilitate generalization. Given a row of the NTK $\Theta_i = \Theta(x_i, \cdot) \in \mathbb{R}^{N_d}$, we define our signal

to be the sum of off-diagonal terms in this row that share a label with $x_i$:

$$S_i = \sum_{\substack{j \neq i \\ y_j = y_i}} \Theta(x_i, x_j)$$

while the corresponding noise measure is simply

$$N_i = \|\Theta_i\|_1 - S_i.$$

The idea behind this metric is that the fitting error at some $\zeta(x_j)$ with $y_i = y_j$ will be closer on average to $\zeta(x_i)$ than $\zeta(x_j)$ such that $y_j \neq y_i$. If $x_i$ is not part of the training set, $\frac{\partial \zeta(x_i)}{\partial t} = -\frac{1}{N_d}\sum_{j=1, j \neq i}^{N_d} \Theta(x_i, x_j)\zeta(x_j)$. Thus if the elements of $\Theta$ with the same label as $x_i$ are large and positive there will be a large magnitude contribution to $\frac{\partial \zeta(x_i)}{\partial t}$ that has the opposite sign as $\zeta(x_i)$ and thus $\zeta(x_i)$ will decrease quickly over time. The noise in this case is the size of the other entries. Generalization error should thus improve if the signal-to-noise ratio

$$\text{SNR} = \frac{1}{N_d}\sum_i \frac{S_i}{N_i} \tag{60}$$

is large and

$$S = \frac{1}{N_d}\sum_i S_i \tag{61}$$

is large as well. The latter condition is important since in the case of networks with small weight variance SNR may be large but $S$ itself vanishes and so will any change in the generalization error. For both networks with $\tanh$ and quantized activatsion we observe that the regime where SNR and $S$ are both large corresponds to the one where the signal propagation time scale in eq. 7 is large as well, as shown in Figure 13.

In this experiment, the network architecture is given by 1 with $L = 30$ and all hidden layers of width 300. Note that for a finite width network with constant layer widths the difference between the NTK and that of a network given by 53 will be a constant factor. The quantities in the plot are averaged over 450 MNIST data points for the $\tanh$ network and 200 images for the quantized network, and 5 different initializations. The NTK for the network with quantized activations is calculated by replacing the terms in the backwards pass with the STE equivalents, as in 57. We note that a similar degradation in the generalization ability when the signal propagation conditions are not satisfied has been described previously in the case of wide networks where only the last layer is trained [15].

### M.5  Change of asymptotic NTK during training

We have argued above that based on the structure of the NTK at initialization for networks where the covariance map has converged, we expect no initial improvement in the generalization error. At later times, if we assume that the Taylor expansion of $\Theta_t^*$ exists

$$\Theta_t^*(z, x') = \sum_{i=0}^{\infty} \frac{t^k}{k!} \frac{\partial^k \Theta_0^*(z, x')}{\partial t^k}$$

we can see directly that $\Theta_t^*(z, x')$ will be independent of $z$ as well, since the summands in the RHS are. This argument thus extends to later times asymptotically at the infinite width limit, or for finite width until such time as deviations from the asymptotic form of the NTK influence the dynamics.

Figure 13: Off-diagonal structure in the NTK is correlated with signal propagation. The signal (eq. 61) that is expected to improve generalization, the signal-to-noise ratio (eq. 60) and the signal propagation time scale (eq. 7) are plotted for different architectures. All quantities are normalized by the maximal value in the range of parameters shown. *Left:* For networks with $\tanh$ activations with different weight variance $\sigma_w^2$, the time scale $\xi$ behaves non-monotonically. The SNR decreases monotonically, while the signal $S$ spikes around the same value of $\sigma_w^2$ where signal propagation is best achieved. Thus the point that maximizes both SNR and $S$ is close to the one where signal propagation is also maximal. *Right:* For networks with quantized activations, as the quantization level increases so does the SNR and the signal itself. We also observe the same non-monotonic behaviour based on the parity of the number of states in all three.

## Footnotes

[5]This can be generalized to other loss functions [16].

[6] Aside from some trivial cases such as learning a constant function.