[Reviews · NeurIPS 2019]

Reviewer 1



ORIGINALITY, QUALITY, CLARITY, and SIGNIFICANCE Originality: To my knowledge, mean field theory for information flow in a neural network with quantized activations has not yet been studied. It is a very reasonable thing to consider. Quality: The theoretical arguments provided by the authors are sound and the numerics are convincing. Clarity: The paper is clearly written. It is apparent that the authors took time to present the main body of the text is a transparent fashion. Significance: On the theory side, this work is a useful addition to understanding the mean field theory of wide networks with quantized activations. I can imagine it being extended to quantized weights as well. DESCRIPTION OF MAIN RESULTS Associated to two inputs x, x’ to a fixed very wide network N, is the empirical variance Q_\ell(x) = \frac{1}{N_\ell}\sum_{i=1}^{N_\ell} [\alpha_i^{(\ell)}(x)]^2 of the size of the vector of pre-activations at input x at layer \ell and the empirical correlation C_\ell(x,x’) = \frac{1}{N_\ell}\sum_{i=1}^{N_\ell} <\alpha_i^{(\ell)}(x), \alpha_i^{(\ell)}(x’)> / [Q_\ell(x) Q_\ell(x’)]^{1/2} In the mean field approximation, these empirical covariances are equal to their statistical averages, and simple dynamical system M(Q_{\ell-1}, C_{\ell-1}) = (Q_\ell, C_\ell) describes their evolution through the network. The authors recall in Lemma 1 the specific form of these dynamics. The authors argue that: (i) the rate of convergence of this dynamical system to a fixed points (Q_*, C_*) is slowest in the C_* direction (see comments below) (ii) once the system has converged to this fixed point, no learning is essentially impossible (see comments below) They use (i) to reduce to study of the full system M to the subsystem M(C):=M(Q_*, C), and their main first result from my point of view (equations (11),(12),(14),(15)) is a formula for M and the rate \chi = dM / dC |_{C*} of convergence to a fixed point C_* for a general family of weakly non-decreasing step function activations \phi_N with N steps defined in (13). This result is exact, given the mean field assumptions. Although the result follows from Lemma 1 by a simple derivation, I still find the final formulas to be quite instructive. The authors illustrate the utility of their formulas is in their second main result, which is a numerical analysis of \chi_N when the steps of \phi_N are equally spaced. The results, illustrated in Figure 2, are quite convincing and suggest that \chi_N grows approximately like N^{1.82}. I also want to mention the authors treatment in appendix D of mean field theory for stochastic rounding for the sign activation. It is a bit more technically involved than the analysis of the deterministic quantized forward pass but is useful since it opens the way for an analysis in appendix L of the NTK. The key point here is that the NTK is a distribution rather than a smooth function a priori for quantized activations but can be turned in a smooth function by annealing over the stochasticity in the rounding procedure. COMMENTS I have a number of comments on the article. Let me begin with what I consider to be the major ones: (1) Around line 87, the authors refer to [25] to justify that the convergence to a fixed point (Q_*, C_*) of M is slowest in the C_* direction. I had a look at [25] and it seems that there is only empirical rather than analytic evidence for this there. If that is right, then I would suggest the authors clarify this point, since it is a key piece of the reasoning. (2) Throughout the article, the authors consider the possibility only of a C_* that lies in [0,1]. It is not a priori clear to me that there is no stable fixed point in [-1,0]. It would have helped me as a reader if the authors remarked on this point. (3) In equation (9), controlling the mean of the average of the singular values m_{JJ^T} of the state-to-state jacobian is often note enough to prevent exploding/vanishing gradients. A number of articles have gone beyond this control either the fluctuations of the average of the singular values or all the singular values (previous articles called this dynamical isometry). I am not suggesting that the authors should do this in the present work since as they point out the J’s don’t really make sense in the quantized setting. But when the authors treat the annealed stochastic rounding version of the NTK, and so it would make sense to say what the authors do prove about the corresponding J_{STE}’s. (4) On line 160, I do not agree with the statement that \chi is bounded above by 2 / \pi. It seems like it is bounded below (!) by 2 / \pi when \sigma_b = 0. Indeed, in this case \chi = (2 / \pi) * 1 / \sqrt{1 - C^2}. Now \sqrt{1 - C^2} is bounded above by 1 on [0,1] and hence 1/ \sqrt{1 - C^2} is bounded below by 1 on [0,1]. Now it is still conceivable that the maximal value of \chi is bounded above by 2 / \pi if the only fixed point is C_* = 0. There are also a few minor points: (1) On line 30 it is not clear what “the signal propagation conditions” means (2) On line 69, it might be good to indicate the the weights and biases are independent. (3) On lines 157-158, I am confused the reasoning for why M cannot have a fixed point. It seems to me that if we consider a model case y(x) = x^2/4 + x/2 + 1/4, then y is convex, differentiable, intersects the line y = x at (1,1) and has slope 1 at the intersection.

Reviewer 2



This paper studies feedforward neural networks with quantized activation function by using MFT, deriving the characteristic time scale \xi which is preferred to be large because it directly determines the trainable layer depth. The result shows that \xi strongly depends on the quantization level N and diverges only in the infinite levels of quantization (N: infinity) or continuum limit. An explicit condition to maximize \xi given quantization level N is derived, yielding an interesting scaling \xi~N^{1.82} for large N. This maximization condition can be used to actually initialize the quantized networks and will be also useful for practical use. For computing the backward gradient, the authors use the straight through estimators to overcome the analytically bad behavior of the activation function. These theoretical predictions are checked by numerical experiments and the theoretical prediction on the trainable layer depth is well confirmed. The authors also discuss about the test accuracy dynamics from a viewpoint of Neural tangent kernel (NTK) or neural network Gaussian process (NNGP) in Section 3.2 and Appendix L, concluding the sufficiently deep networks with sufficiently small learning rate and without critical initialization cannot generalize in principle, because the kernel converges at deep layers to a trivial form which cannot discriminate any test points. This point has, however, been already discussed in arXiv:1711.00165 titled Deep neural networks as gaussian processes'' and more detailed numerical experiments have been given there. Hence I think the authors should cite this reference. Even with this overlap with the preceding study, the main claim of this paper about the quantization effect in activation function is surely new and important, and thus I think there is no need of large modification in the manuscript. I have one concerning point in the main text. For the experiment described in lines 208-217, I cannot find the result, though I may misunderstand something. Please clarify this point. Originality: Application of MFT to studying the effect of quantization is clearly new and important. Quality: The submission is technically sound and the theoretical prediction is well supported by experiments. The quality is high enough. Clarity: The paper is well organized and equipped with nice appendices well summarizing the detailed computations. No need of large modifications. I only give the list of typos and uneasy-to-understand expressions which I found: Line 114: gradient form -> gradient from Line 215: We than -> We then Line 574: do not obtain -> are not satisfied Lines 578-579: Should explain what is N_d Lines 621-623: I could not understand the manipulations here. Could you explain them in more detail? Line 660: phenonmenon -> phenomenon Significance: I think this work is an important one because it provides a theoretical knowledge about the quantization-depth tradeoff and the best'' initialization method under a given quantization level which is also useful for practitioners. These are firm theoretical and methodological advance which are never known.

Reviewer 3



Originality Although the theoretical framework and method in the paper are not new, the paper derives novel theoretical results and supports them with empirical experiments. Related work is clearly cited and discussed, and the authors make clear how the current work differs from previous ones. Quality and clarity The work seems to me sound, with adequate theoretical and empirical analyses. The paper is clear and very readable, and I appreciate the choice of the authors to clearly present and elaborate on the approach in section 3, despite space limitation. I have only text-correction comments, - typo in footnote 1 ('under') - typo in line 179. - Figure 1, consider adding a scale and move it to eg page 3. - Ref [14] is not complete. - type in line 242: 'show' --> 'shows' Significance I find that the main contribution of the paper is the derivation of a relation between the maximal trainable depth of a network and the desired quantization level, which was further supported by simulations. This seems an important result for future studies that aim to train deep networks in low-resource setups.

[Author Response · NeurIPS 2019]

We thank the reviewers for their positive opinions and constructive feedback. Responses are given below. All the issues discussed (and the typos) will be fixed in the final version of the paper. In addition, since submission, we found that our main results can be straightforwardly extended to provide an upper bound on the timescale/lengthscale of signal propagation in quantized RNNs/CNN. This is done by a combination of our results with previous mean-field results on RNNs/CNNs. We plan to add this practically relevant extension to the main paper as a short paragraph. We respond to specific comments below:

**R1: "...the authors refer to [25] to justify that the convergence to a fixed point $(Q^*, C^*)$ of M is slowest in the $C^*$ direction ... it seems that there is only empirical rather than analytic evidence for this..."** » Our original intention was to reference the analytic claim in [25], which appears their on the second paragraph of page 5. However, upon closer inspection, we now see that their analysis is inapplicable for quantized activations (as it is based on a Taylor-expansion of the activation function, done in appendix 7.1 there). We will therefore clarify that this claim is currently only empirical. We will also add our own empirical evidence, which supports this claim for quantized activations. It might be possible also to get an analytical argument for quantized activations, but we are not sure yet.

**R1: "Throughout the article, the authors consider the possibility only of a $C^*$ that lies in [0,1]. It is not a priori clear to me that there is no stable fixed point in [-1,0]..."** » Indeed, this should have been better clarified. We will add a short description to address the region [-1, 0). In general, our equations do hold for this region. When there is bias ($\sigma_b^2 > 0$) there will be no fixed point in [-1,0), because the bias makes the hidden states more positively correlated while the rest of the operations bring the correlation closer to zero. In the case of $\sigma_b^2 = 0$, assuming an anti-symmetric activation function (as was used for activation with constant/linear spacing), the entire network becomes anti-symmetric upon initialization and $C = -1$ becomes an infinitely unstable fixed point as well.

**R1: (point 3)"... But when the authors treat the annealed stochastic rounding version of the NTK, and so it would make sense to say what the authors do prove about the corresponding $J_{STE}$'s"** » As the reviewer points out, in the stochastic rounding setting one could study the moments of the spectrum of $J_{STE}$ and obtain dynamical isometry conditions. Indeed, this is an interesting topic for future work. We will clarify that we have not explored it in the present work.

**R1: "On line 160, I do not agree with the statement that $\chi$ is bounded above by $\frac{2}{\pi}$. It seems like it is bounded below (!) by $\frac{2}{\pi}$ when $\sigma_b = 0$..."** » Please see the proof in appendix K.1 which fully covers this issue: it shows that increasing $\sigma_b$ will, in fact, result in a smaller slope $\chi$ at the fixed point. If this was not sufficiently clear from the main text, we can also give the reader some intuition about this: by adding the approximated equation for the fixed point slope using a Taylor expansion $\left( C^* \simeq 1 - \left(\frac{8}{\pi^2}\right) \left(\frac{\sigma_w^2}{\sigma_w^2 + \sigma_b^2}\right)^2 \right)$, and plugging it into equation (11): $\chi = \frac{2\sigma_w^2}{\pi(\sigma_w^2 + \sigma_b^2)\sqrt{1 - (C^*)^2}}$, we obtain an expression decreasing in $\sigma_b^2$. We ultimately dropped this argument from the final version, after we derived the proof in appendix K.1. However, we can add this argument back if it is helpful.

**R1: "On lines 157-158, I am confused the reasoning for why M cannot have a fixed point..."** » Please note the additional condition, regarding $\mathcal{M}(C)$ diverging at $C = 1$, forcing $\mathcal{M}(1 - \epsilon) < 1 - \epsilon$ for some $\epsilon > 0$. With $\mathcal{M}(0) \geq 0$ and $\mathcal{M}(C)$ being convex, the fixed point slope can not exceed the slope of the linear function between those points: $\frac{\mathcal{M}(1-\epsilon) - \mathcal{M}(0)}{1-\epsilon} < \frac{1-\epsilon}{1-\epsilon} = 1$.

**R2: "This point [about the test accuracy dynamics from a viewpoint of Neural tangent kernel (NTK)] has, however, been already discussed in arXiv:1711.00165 titled "Deep neural networks as gaussian processes" and more detailed numerical experiments have been given there"** » The covariance map studied in this part of our work is indeed identical to the NNGP kernel in arXiv:1711.00165 which can be used for inference. However the authors of that work do not consider inference with the NTK or the dynamics of the generalization error of neural networks in the linear regime. Their argument in section 3 is indeed similar to ours, but applies to NNGP inference (rather than wide neural networks trained with gradient descent). We thank the reviewer for pointing this out, and we will add a reference and discussion in the revision.

**R2: "... For the experiment described in lines 208-217, I cannot find the result"** » The relevant figure for this experiment is figure 1, but the reference is missing. Will fix.

**R2: "Lines 621-623: I could not understand the manipulations here. Could you explain them in more detail?"** » $\gamma$ is chosen as a smooth extension of $\widehat{\alpha}$ in order to enable one to take derivatives in the lines below. The second order terms drop from symmetry. This calculation is a recapitulation of the one in [13]. We will clarify this further.

[Meta-Review · NeurIPS 2019]

The paper provides a mean-field analysis of infinitely wide neural networks with quantized activations, proposing a relation between the choice of initialization hyper-parameters and the maximal depth by primarily by considering how correlations between two inputs propagate through the network at initialization as well as numerical stability issues. All reviewers agree that it is a good contribution.